# Robustness to Adversarial Perturbations in Learning from Incomplete Data

**Amir Najafi**
Department of Computer Engineering
Sharif University of Technology
Tehran, Iran
najafy@ce.sharif.edu

**Shin-ichi Maeda**
Preferred Networks, Inc.
Tokyo, Japan
ichi@preferred.jp

**Masanori Koyama**
Preferred Networks, Inc.
Tokyo, Japan
masomatics@preferred.jp

**Takeru Miyato**
Preferred Networks, Inc.
Tokyo, Japan
miyato@preferred.jp

## Abstract

What is the role of unlabeled data in an inference problem, when the presumed underlying distribution is adversarially perturbed? To provide a concrete answer to this question, this paper unifies two major learning frameworks: Semi-Supervised Learning (SSL) and Distributionally Robust Learning (DRL). We develop a generalization theory for our framework based on a number of novel complexity measures, such as an adversarial extension of Rademacher complexity and its semi-supervised analogue. Moreover, our analysis is able to quantify the role of unlabeled data in the generalization under a more general condition compared to the existing theoretical works in SSL. Based on our framework, we also present a hybrid of DRL and EM algorithms that has a guaranteed convergence rate. When implemented with deep neural networks, our method shows a comparable performance to those of the state-of-the-art on a number of real-world benchmark datasets.

## 1 Introduction

Robustness to adversarial attacks is an essential feature in the design of modern classifiers —in particular, of deep neural networks [1, 2]. Adversarial Training (AT) [3], Virtual AT [4] and Distillation [5] are examples of promising approaches to defend against a *point-wise adversary* who can alter input data-points in a separate manner. However, as shown by [6], a good defense against a *distributional adversary* who shifts the input distribution instead of data-points could improve the robustness of a classifier more effectively. This has led to the development of Distributionally Robust Learning (DRL) [7], which has recently attracted intensive research interest [8, 9, 10, 11]. Despite of all the advancements in supervised DRL, the number of studies that tackle this problem from a semi-supervised angle is slim to none [12]. Motivated by this fact, we propose a distributionally robust framework that handles Semi-Supervised Learning (SSL) scenarios. Our work is an extension of self-learning [13, 14, 15], which encompasses methods such as Expectation-Maximization (EM) algorithm. It can also cope with any existing classifier such as neural networks. Intuitively, we first infer soft-labels for the unlabeled data, and then search for suitable classification rules that show low sensitivity to adversarial perturbation around these soft-label distributions.

Parts of this paper can be considered as a semi-supervised extension of [9]. Computational complexity of our framework is comparable to those of its supervised rivals. Moreover, we design a Stochastic Gradient Descent (SGD)-based algorithm with a guaranteed convergence rate to optimize our model.

|  | DRL | PL | VAT | SSDRL |
|---|:---:|:---:|:---:|:---:|
| Generalization Bound | ✓ | × | × | ✓ |
| Convergence Guarantee | ✓ | × | × | ✓ |
| Adversarial Robustness | ✓ | × | ✓ | ✓ |
| Semi-Supervised Learning | × | ✓ | ✓ | ✓ |

Table 1: Comparison between the proposed framework (SSDRL) and some existing methods: DRL of [9], Pseudo Labeling (PL) [19], and Virtual Adversarial Training (VAT) [4].

In order to address the generalization, we introduce a set of novel complexity measures such as *Adversarial Rademacher Complexity* and *Minimum Supervision Ratio* (MSR), which are based on the hypothesis set and input data distribution. We show that if the ratio of the labeled samples in a dataset (supervision ratio) exceeds MSR, true adversarial risk can be bounded. Also, proper parameter adjustment can arbitrarily decrease MSR at the cost of increasing the generalization bound; This means our theoretical guarantees hold for all supervision ratios. The theoretical contribution of our work is summarized in Table 1. We have also tested our method, denoted by SSDRL, via extensive computer experiments on datasets such as MNIST [16], SVHN [17], and CIFAR-10 [18]. When equipped with deep neural networks, SSDRL outperforms rivals such as Pseudo-Labeling (PL) [19] and the supervised DRL of [9] on all the datasets. Also, SSDRL outperforms VAT [4] on SVHN, while it demonstrates a comparable performance on MNIST and CIFAR-10.

The rest of the paper is organized as follows: Section 1.1 specifies the notations, and Section 1.2 reviews the related works. The proposed framework is presented in Section 2, where numerical optimization is explained in Section 2.1 and its generalization is addressed in Section 2.2. Section 3 is devoted to experimental results. Finally, Section 4 concludes the paper.

## 1.1 Notations

We extend the notations used in [9]. Assume $\mathcal{Z}$ to be an input space, $\Theta$ to be a parameter set, and $\ell : \mathcal{Z} \times \Theta \to \mathbb{R}$ a parametric loss function. Observation space $\mathcal{Z}$ can either be the feature space $\mathcal{X}$ in unsupervised scenarios, or the space of feature-label pairs, i.e., $\mathcal{Z} \triangleq \mathcal{X} \times \mathcal{Y}$, where $\mathcal{Y}$ denotes the set of labels. For simplicity, we only consider finite label-sets. By $M(\mathcal{Z})$, we mean the set of all probability measures supported on $\mathcal{Z}$. Let us denote the function $c : \mathcal{Z} \times \mathcal{Z} \to [0, +\infty)$ as the *transportation cost*. Under some conditions on $c$, Definition B.1 (supplementary) formulates the *Wasserstein* distance $W_c(P, Q)$ between two distributions $P, Q \in M(\mathcal{Z})$, w.r.t. $c$ [8]. $W_c(P, Q)$ measures the minimal cost of moving $P$ to $Q$, where the cost of moving one unit of mass from $z$ to $z'$ is given by $c(z, z')$. Also, for $\epsilon \geq 0$ and a distribution $Q \in M(\mathcal{Z})$, we define an $\epsilon$-ambiguity set as $\mathcal{B}_\epsilon(Q) \triangleq \{P \in M(\mathcal{Z}) \mid W_c(P, Q) \leq \epsilon\}$. Training dataset is shown by $D \triangleq \{Z_1, \ldots, Z_n\}$, which includes i.i.d. samples drawn from a fixed (and unknown) distribution $P_0 \in M(\mathcal{Z})$, while $n$ denotes the dataset size. For a dataset $D$, let $\hat{\mathbb{P}}_D \in M(\mathcal{Z})$ be $\hat{\mathbb{P}}_D \triangleq \frac{1}{n} \sum_{i=1}^n \delta_{Z_i}$, where $\delta_z$ denotes the Dirac delta function at point $z \in \mathcal{Z}$. Accordingly, $\mathbb{E}$ and $\hat{\mathbb{E}}_D$ represent the statistical and empirical expectation operators, respectively. For a distribution $P \in M(\mathcal{X} \times \mathcal{Y})$, $P_X$ denotes the marginal distribution $P_X(\cdot) \triangleq \sum_{y \in \mathcal{Y}} P(\cdot, y)$ over $\mathcal{X}$, and $P_{|X} \in M(\mathcal{Y})$ is the conditional distributions over labels given feature vector $X \in \mathcal{X}$. To simplify the notations, for $z = (X, y) \in \mathcal{X} \times \mathcal{Y}$ and a function $f$, the notations $f(z)$ and $f(X, y)$ have been used, interchangeably.

## 1.2 Background and Related Works

DRL minimizes a worst-case risk against an adversary, who has a limited budget to alter the data distribution $Q \in M(\mathcal{Z})$ in order to inflict the maximum possible damage. Here, $Q$ can either be the true measure $P_0$, or the empirical one $\hat{\mathbb{P}}_D$ [10]. Mathematically, DRL is formulated as [8, 11]:

$$\inf_{\theta \in \Theta} \sup_{P \in \mathcal{B}_\epsilon(Q)} \mathbb{E}_P \{\ell(Z; \theta)\}. \tag{1}$$

*Wasserstein* metric has been widely used to quantify the strength of adversarial attacks [8, 9, 11, 12], thanks to (i) its relations to adversarial robustness [20] and (ii) suitable dual-form properties [11]. In [8], authors have reformulated DRL into a convex program for Logistic Regression. Convergence and generalization of DRL, in a general context, have been addressed in [9], while adjusting the ambiguity

set size, i.e. $\epsilon$, has been tackled in [21]. In [22], the authors have investigated the convergence of the inner maximization of (1), and its effects on the later stages of adversarial training. An analysis on DRL methods with $f$-divergences is given in [10]. Also, sample complexity of DRL has been reviewed by [23] and [24].

Abundance of unlabeled data has made SSL methods widely popular [4, 25]. See [14] for a review on classical SSL approaches. Many robust SSL algorithms have been proposed so far [26, 27], however, their notion of robustness is different from the one considered here. More recent works include [28, 29, 30, 31]. In [29], [30] and [31], authors have mainly focused on a Gaussian model for theoretical validation, while [28] empirically shows that robust self-training eliminates the accuracy-robustness tradeoff by leveraging unlabeled data. In [32], a pessimistic SSL approach is proposed that provably enhances the performance by incorporating the unlabeled data.We show that a special case of our method reduces to an adversarial extension of [32]. Guarantees on the generalization of SSL can only be made under certain assumptions on the choice of hypothesis set and the true data distribution [14, 15, 33]. For example, a *compatibility* function is introduced in [15] to restrict the relation between the model set and input data distribution. Also, in [34], author has theoretically analyzed SSL under the *cluster assumption*.The main reason for making such assumptions is that lack of any knowledge on the relation of a feature vector to its label, simply makes unlabeled data useless for classification. In Section 2.2, we propose a novel compatibility criterion under a general setting which enables us to establish a generalization theory for our work.

Finally, the only existing work that also falls in the cross section of DRL and SSL is [12]. However, this method severely restricts the shifted distributions, so that the adversary can only choose from a set of delta-spikes over labeled and augmented unlabeled samples.

## 2 Proposed Framework

From now on, let $\mathcal{Z} \triangleq \mathcal{X} \times \mathcal{Y}$. In SSL, a dataset $\boldsymbol{D}$ consists of two non-overlapping parts: $\boldsymbol{D}_{\mathrm{l}}$ (labeled) and $\boldsymbol{D}_{\mathrm{ul}}$ (unlabeled). Let us denote $\mathcal{I}_{\mathrm{l}}$ and $\mathcal{I}_{\mathrm{ul}}$ as the index sets corresponding to these parts, respectively. Thus, we have $\boldsymbol{D}_{\mathrm{l}} = \left\{ (\boldsymbol{X}_i, y_i) \,|\, i \in \mathcal{I}_{\mathrm{l}} \right\}$, and $\boldsymbol{D}_{\mathrm{ul}} = \left\{ \boldsymbol{X}_i \,|\, i \in \mathcal{I}_{\mathrm{ul}} \right\}$. The hidden labels in $\boldsymbol{D}_{\mathrm{ul}}$ can be modeled by random variables supported on $\mathcal{Y}$. Note that DRL in (1) cannot readily apply to this *partially-labeled* setting, since (1) needs complete knowledge of all the feature-label pairs in $\boldsymbol{D}$. To overcome this problem, first let us make the following definition:

**Definition 1.** *The consistent set of probability distributions* $\hat{\mathcal{P}}(\boldsymbol{D}) \subseteq M(\mathcal{Z})$ *with respect to a partially-labeled dataset* $\boldsymbol{D} = \boldsymbol{D}_{\mathrm{l}} \cup \boldsymbol{D}_{\mathrm{ul}}$ *is defined as*

$$\hat{\mathcal{P}}(\boldsymbol{D}) \triangleq \left\{ \left( \frac{n_{\mathrm{l}}}{n} \right) \hat{\mathbb{P}}_{\boldsymbol{D}_{\mathrm{l}}} + \left( \frac{n_{\mathrm{ul}}}{n} \right) \hat{\mathbb{P}}_{\boldsymbol{D}_{\mathrm{ul}}} \cdot \Omega \,\big|\, \Omega \in M^{\mathcal{X}}(\mathcal{Y}) \right\},$$

*where* $n_{\mathrm{l}}$ *and* $n_{\mathrm{ul}}$ *(with* $n = n_{\mathrm{l}} + n_{\mathrm{ul}}$*) are the sizes of* $\boldsymbol{D}_{\mathrm{l}}$ *and* $\boldsymbol{D}_{\mathrm{ul}}$*, respectively, and* $M^{\mathcal{X}}(\mathcal{Y})$ *denotes the set of all conditional distributions supported on* $\mathcal{Y}$*, given features in* $\mathcal{X}$*.*

All possible (soft-)labelings of unlabeled samples in $\boldsymbol{D}_{\mathrm{ul}}$ are collected in $\hat{\mathcal{P}}(\boldsymbol{D})$. Note that the empirical measure corresponding to the true complete dataset is also included in the consistent set. Our aim is to choose a suitable measure from this set, and then use it for (1).

We take a known family of SSL approaches, called self-learning [13, 35], and then combine it with DRL. Self-learning methods, e.g. EM algorithm [36], transfer the knowledge from labeled samples to unlabeled ones through *pseudo-labeling*. More precisely, a learner is trained on the supervised portion of a dataset, and then employs its learned rules to assign pseudo-labels to the remaining unlabeled part. However, such methods are prone to over-fitting if the information flow from $\boldsymbol{D}_{\mathrm{l}}$ to $\boldsymbol{D}_{\mathrm{ul}}$ is not properly controlled. One way to overcome this issue is to use soft-labeling, which maintains a minimum level of uncertainty over unlabeled samples. By combining the above arguments with the core idea of DRL in (1), we propose the following learning scheme:

$$\inf_{\theta \in \Theta} \ \inf_{S \in \hat{\mathcal{P}}(\boldsymbol{D})} \left\{ \sup_{P \in \mathcal{B}_{\epsilon}(S)} \mathbb{E}_P \left\{ \ell(\boldsymbol{X}, y; \theta) \right\} + \left( \frac{1-\eta}{\lambda} \right) \hat{\mathbb{E}}_{\boldsymbol{D}_{\mathrm{ul}}} \left\{ \mathbb{H}\left( S_{|\boldsymbol{X}} \right) \right\} \right\}, \qquad (2)$$

where $\lambda$ is a user-defined parameter, $\eta \triangleq n_{\mathrm{l}}/n$ is the *supervision ratio*, and $\mathbb{H}(\cdot)$ denotes the Shannon entropy. For now, let us assume $\lambda < 0$.

Minimization over $S \in \hat{\mathcal{P}}(\boldsymbol{D})$ acts as a *knowledge transfer* module that finds the optimal distribution in $\hat{\mathcal{P}}(\boldsymbol{D})$. Again, note that distributions in $\hat{\mathcal{P}}(\boldsymbol{D})$ vary only in the way they assign (soft-)labels to unlabeled data. The scheme in (2) is based on *optimism* in the sense that, for any $\theta \in \Theta$, learner is instructed to pick the labels that are more likely to reduce the average loss function $\ell(\cdot; \theta)$ for each unlabeled sample. This is the core idea of self-learning. However, a pessimistic learner does the opposite, i.e. picks the less likely labels with large loss values and hence does not trust the loss function. The negative regularization term $\frac{1-\eta}{\lambda}\hat{\mathbb{E}}_{\boldsymbol{D}_{\mathrm{ul}}}\left\{\mathbb{H}\left(S_{|\boldsymbol{X}}\right)\right\}$ prevents hard decisions for labels and promotes soft-labeling by bounding the Shannon entropy of label-conditionals from below. A smaller $|\lambda|$ gives softer labels. In the extreme case, choosing $\lambda = -\infty$ ends up in an adversarial version of the self-training in [14]. It should be noted that according to (2), learner is forced to show less sensitivity near all (labeled and unlabeled) training data, just as one expects from a semi-supervised DRL.

We show that (2) can be efficiently solved given that some smoothness conditions hold for $\ell$ and $c$. Before that, Theorem 1 shows that the optimization corresponding to the knowledge transfer module has an analytic solution, which implies the computational cost of (2) is only slightly higher than those of its fully-supervised counterparts, such as [9].

**Theorem 1** (Lagrangian-Relaxation). *For any continuous loss $\ell : \mathcal{Z} \times \Theta \to \mathbb{R}$ and $c : \mathcal{Z} \times \mathcal{Z} \to \mathbb{R}_{\geq 0}$, parameters $\epsilon \geq 0$, $\gamma \geq 0$ and $\lambda \in \mathbb{R} \cup \{\pm\infty\}$, and a partially-labeled dataset $\boldsymbol{D}$ with size $n$, let us define the empirical Semi-Supervised Adversarial Risk (SSAR), denoted by $\hat{R}_{\mathrm{SSAR}}(\theta; \boldsymbol{D})$, as*

$$\hat{R}_{\mathrm{SSAR}}(\theta; \boldsymbol{D}) \triangleq \frac{1}{n}\sum_{i \in \mathcal{I}_{\mathrm{l}}}\phi_{\gamma}(\boldsymbol{X}_i, y_i; \theta) + \frac{1}{n}\sum_{i \in \mathcal{I}_{\mathrm{ul}}}\operatorname*{softmin}_{y \in \mathcal{Y}}^{(\lambda)}\{\phi_{\gamma}(\boldsymbol{X}_i, y; \theta)\} + \gamma\epsilon, \qquad (3)$$

*where the adversarial loss $\phi_{\gamma}(\boldsymbol{X}, y; \theta)$, and the soft-minimum operator $\operatorname{softmin}_{y \in \mathcal{Y}}^{(\lambda)}(\boldsymbol{q})$ for any $\boldsymbol{q} \in \mathbb{R}^{\mathcal{Y}}$ are defined as*

$$\phi_{\gamma}(\boldsymbol{X}, y; \theta) \triangleq \sup_{\boldsymbol{z}' \in \mathcal{Z}}\ell(\boldsymbol{z}'; \theta) - \gamma c(\boldsymbol{z}', (\boldsymbol{X}, y)), \text{ and } \operatorname*{softmin}_{y \in \mathcal{Y}}^{(\lambda)}(\boldsymbol{q}) \triangleq \frac{1}{\lambda}\log\left(\frac{1}{|\mathcal{Y}|}\sum_{y \in \mathcal{Y}}e^{\lambda q_y}\right), \quad (4)$$

*respectively. Let $\theta^* \in \Theta$ be a minimizer of (2) for some given $\epsilon \geq 0$ and $\lambda < 0$. Then, there exists $\gamma \geq 0$ such that $\theta^*$ is also a minimizer of (3) with the same parameter setting.*

Proof of Theorem 1 is given in Appendix D. Note that $\operatorname{softmin}$ equals to : (i) $\min$ operator for $\lambda = -\infty$, (ii) average for $\lambda = 0$, and (iii) $\max$ for $\lambda = +\infty$. Also, $\epsilon$ and $\gamma$ are non-negative dual parameters and fixing either of them uniquely determines the other one. Therefore, one can adjust $\gamma$ (for example via cross-validation), instead of $\epsilon$. See [9] for a similar discussion.

A more subtle look at (3) shows that in the dual context of the proposed scheme, one is free to also consider positive values for $\lambda$. The sign of $\lambda$ indicates *optimism* ($\lambda \leq 0$), or *pessimism* ($\lambda > 0$) during the (soft-)label assignment. The choice between optimism vs. pessimism depends on the *compatibility* of the model set $\Theta$ with the true distribution $P_0$. In Section 2.2, we show that enabling $\lambda$ to take values in $\mathbb{R}$ rather than $\mathbb{R}^-$ is crucial for establishing a generalization bound for (3). In other words, for a *very bad* hypothesis set, one must choose to be pessimistic to be able to generalize well. To see situations where pessimism in SSL helps, reader can refer to [32].

## 2.1 Numerical Optimization

We propose a numerical optimization scheme for solving (3) which has a convergence guarantee. Lemmas E.1 and E.2 (supplementary) explicitly compute the gradients of (3). This way, one can simply apply the mini-batch SGD to solve for (2) via Algorithm 1. Note that due to the strong concavity property in Lemma E.1, $\delta$ can be chosen arbitrarily small. Other parameters such as $\gamma$ (or equivalently $\epsilon$) and $\lambda$ should be adjusted via cross-validation. The computational complexity of Algorithm 1 is no more than $\eta + |\mathcal{Y}|(1 - \eta)$ times of that of [9], where the latter can only handle supervised data[1]. Note that Algorithm 1 reduces to [9] in fully-supervised scenarios, and coincides with Pseudo-Labeling and EM algorithm when $(\gamma = \infty, \lambda = -\infty)$ and $(\gamma = \infty, \lambda = -1)$,

**Algorithm 1** Stochastic Gradient Descent for SSDRL

---

1: Inputs: $\boldsymbol{D}, \gamma, \lambda, (k \leq n, \delta, \alpha, T)$
2: Initialize $\theta_0 \in \Theta$, and set $t \leftarrow 0$.

3: **for** $t = 0 \rightarrow T - 1$ **do**
4:      Randomly select index set $\mathcal{I} \subseteq [n]$ with size $k$.
5:      **for** $i \in \mathcal{I}_l \cap \mathcal{I}$ **do**
6:          Compute a $\delta$-approx of $\boldsymbol{z}_i^* (\theta_t)$ from Lemma E.2.
7:      **end for**
8:      **for** $(i, y) \in (\mathcal{I}_{\mathrm{ul}} \cap \mathcal{I}) \times \mathcal{Y}$ **do**
9:          Compute a $\delta$-approximate of $\boldsymbol{z}_i^* (y; \theta_t)$ from Lemma E.2.
10:      **end for**
11:      Compute the sub-gradient of $\hat{R}_{\mathrm{SSAR}} (\theta; \boldsymbol{D})$ from (E.3) (Lemma E.2) at point $\theta = \theta_t$
         using only samples in $\mathcal{I}$, and denote it with $\partial_\theta \hat{R}_{\mathrm{SSAR}} (\theta_t; \boldsymbol{D})$.
12:      $\theta_{t+1} \leftarrow \mathrm{Proj}_\Theta \left( \theta_t - \alpha \partial_\theta \hat{R}_{\mathrm{SSAR}} (\theta_t; \boldsymbol{D}) \right)$
13: **end for**
14: Output: $\theta^* \leftarrow \theta_T$

---

respectively. The following theorem guarantees the convergence of Algorithm 1 to a local minimizer of (3).

**Theorem 2.** *Assume loss function $\ell$, transportation cost $c$, $\gamma \geq 0$ and $|\lambda| < \infty$ satisfy the conditions of Lemma E.2. Also, assume $\ell$ is differentiable w.r.t. both $\boldsymbol{z}$ and $\theta$, with Lipschitz gradients. Also, let $\|\nabla_\theta \ell (\boldsymbol{z}; \theta)\|_2 \leq \sigma$ for some $\sigma \geq 0$ all over $\mathcal{Z} \times \Theta$. Denote $\theta_0 \in \Theta$ to be an initial hypothesis, and $\theta^* \in \Theta$ as a local minimizer of (3). Assume the partially-labeled dataset $\boldsymbol{D}$ includes $n$ i.i.d. training samples. Also, let $\Delta \hat{R} \triangleq \hat{R}_{\mathrm{SSAR}} (\theta_0; \boldsymbol{D}) - \hat{R}_{\mathrm{SSAR}} (\theta^*; \boldsymbol{D})$. Then, for a fixed step size $\alpha^*$, the outputs of Algorithm 1 with parameters $k = 1$, $\delta > 0$, $\alpha = \alpha^*$ after $T$ iterations, say $\theta_1, \ldots, \theta_T$, satisfy the following inequality:*

$$\frac{1}{T} \sum_{t=1}^{T} \mathbb{E} \left\{ \left\| \nabla_\theta \hat{R}_{\mathrm{SSAR}} (\theta_t; \boldsymbol{D}) \right\|_2^2 \right\} \leq 4\sigma^2 \sqrt{\frac{\Delta \hat{R}}{T} \left( \frac{B}{\sigma^2} + (1 - \eta) |\lambda| |\mathcal{Y}| \right)} + C\delta, \qquad (5)$$

*where constants $B$ and $C$ and step size $\alpha^*$ only depend on $\gamma$ and Lipschitz constants of $\ell$.*

The proof of Theorem 2 with explicit formulations for constants $B$ and $C$ and step size $\alpha^*$ are given in Appendix D (supplementary). Theorem 2 guarantees a convergence rate of $O\left(T^{-1/2}\right)$ for Algorithm 1, if one neglects $\delta$. Note that the presence of $\delta$ is necessary since one cannot find the exact maximizer of (E.2) in finite steps. However, due to Lemma E.1, $\delta$ can become infinitesimally small. Theorem D.1 (supplementary) guarantees the convergence of Algorithm 1 in hard-decision regimes, i.e. $\lambda = \pm\infty$. Note that $\ell$ is not necessarily convex w.r.t. $\theta$, e.g. neural nets. However, given a convex loss $\ell$, Theorem D.2 gives us a conditions on $\lambda$ to guarantee the convexity of (3) as well.

## 2.2 Generalization Guarantee

We intend to bound the *true adversarial risk*, i.e. $\sup_{P \in \mathcal{B}_\epsilon (P_0)} \mathbb{E}_P \{\ell (\boldsymbol{Z}; \theta^*)\}$, where $\theta^*$ denotes the optimizer of the empirical risk in (3). However, the two major concerns are: (i) we are training our model against an adversary, and (ii) our training dataset is partially labeled. We first address these issues and then present our main contribution in Theorem 3.

Classical Rademacher complexity, denoted by $\mathcal{R}_n (\mathcal{F})$, measures how well a function set $\mathcal{F}$ can learn *noise*, and thus is exposed to over-fitting on small datasets. We give a novel adversarial extension for $\mathcal{R}_n$ which also appears in our generalization bound. Moreover, we show it converges to zero when $n \rightarrow \infty$, for all function sets with a finite VC-dimension. Before that, let us define the set of $\epsilon$-Monge maps as $\mathcal{A}_\epsilon \triangleq \{a : \mathcal{Z} \rightarrow \mathcal{Z} | c (\boldsymbol{z}, a (\boldsymbol{z})) \leq \epsilon, \forall \boldsymbol{z} \in \mathcal{Z}\}$.

**Definition 2** (Semi-Supervised Monge (SSM) Rademacher Complexity). *For $\mathcal{Z} \triangleq \mathcal{X} \times \mathcal{Y}$, assume a function set $\mathcal{F} \subseteq \mathbb{R}^{\mathcal{Z}}$ and a distribution $P_0 \in M(\mathcal{Z})$. For $\epsilon \geq 0$ and $n \in \mathbb{N}$, let us define*

$$g_{\mathrm{l}}(n) \triangleq \mathbb{E}_{\boldsymbol{Z}_{1:n}, \boldsymbol{\sigma}} \left\{ \sup_{f \in \mathcal{F}} \frac{1}{n} \sum_{i=1}^{n} \sigma_i \left[ \sup_{a \in \mathcal{A}_\epsilon} f(a(\boldsymbol{Z}_i)) \right] \right\}$$

$$\text{and} \quad g_{\mathrm{ul}}(n) \triangleq \sum_{y \in \mathcal{Y}} \mathbb{E}_{\boldsymbol{X}_{1:n}, \boldsymbol{\sigma}} \left\{ \sup_{f \in \mathcal{F}} \frac{1}{n} \sum_{i=1}^{n} \sigma_i \left[ \sup_{a \in \mathcal{A}_\epsilon} f(a(\boldsymbol{X}_i, y)) \right] \right\},$$

*where $\boldsymbol{Z}_{1:n} \overset{i.i.d.}{\sim} P_0$ and $\boldsymbol{X}_{1:n} \overset{i.i.d.}{\sim} P_{0\boldsymbol{X}}$. $\boldsymbol{\sigma} \in \{-1, +1\}^n$ indicates a vector of i.i.d. symmetric Rademacher variables. Then, for $\eta \in [0, 1]$, the SSM Rademacher complexity of $\mathcal{F}$ is defined as*

$$\mathcal{R}_{n,(\epsilon,\eta)}^{(\mathrm{SSM})}(\mathcal{F}) \triangleq \eta g_{\mathrm{l}}(\lceil n\eta \rceil) + (1 - \eta) g_{\mathrm{ul}}(\lceil n(1 - \eta) \rceil).$$

By setting $\epsilon = 0$ and $\eta = 1$, the above definition reduces to the classical Rademacher complexity $\mathcal{R}_n$. We define a function set to be *learnable*, if $\lim_{n \to \infty} \mathcal{R}_n = 0$. Similarly, a function class $\mathcal{F}$ is said to be *adversarially learnable* w.r.t. parameters $(\epsilon, \eta)$, if $\lim_{n \to \infty} \mathcal{R}_{n,(\epsilon,\eta)}^{(\mathrm{SSM})}(\mathcal{F}) = 0$. But, how can we numerically compute this measure in practice? The main difference between $\mathcal{R}_n$ and SSM Rademacher complexity is that the latter adversarially alters the input distribution. However, many distribution-free bounds already exist for $\mathcal{R}_n$ [37], which apply to almost all practical function sets, e.g. classifiers with a bounded VC-dimension (e.g. neural nets), restricted regression tools and etc. We show that by having a distribution-free bound on the Rademacher complexity of $\mathcal{F}$, one can also bound the SSM Rademacher complexity. Mathematically speaking, assume that there exists an asymptotically decreasing upper-bound $\Delta(n)$ such that $\mathcal{R}_n(\mathcal{F}) \leq \Delta(n), \; \forall P_0 \in M(\mathcal{X} \times \mathcal{Y})$. Then, for all $\eta \in [0, 1]$ and $\epsilon \geq 0$, we have (Lemma E.3):

$$\mathcal{R}_{n,(\epsilon,\eta)}^{(\mathrm{SSM})}(\mathcal{F}) \leq \eta \Delta(\lceil n\eta \rceil) + (1 - \eta) |\mathcal{Y}| \Delta(\lceil n(1 - \eta) \rceil), \tag{6}$$

where the r.h.s. of (6) goes to zero as $n \to \infty$. This includes almost all practical classifiers, e.g. neural nets, support vector machines, random forests and etc. For example, consider the $0 - 1$ loss for a classifier with a VC-dimension of $\dim(\Theta)$. Then, due to Dudley's entropy bound and Haussler's upper-bound [37], there exists constant $C$ such that, regardless of $\epsilon$ or $P_0$, we have (Lemma E.3):

$$\Delta(n) \leq C\sqrt{\frac{\dim(\Theta)}{n}}, \quad \text{and thus} \quad \mathcal{R}_{n,(\epsilon,\eta)}^{(\mathrm{SSM})}(\mathcal{F}) \leq C\sqrt{\frac{\dim(\Theta)}{n}} \left(\sqrt{\eta} + \sqrt{1 - \eta} |\mathcal{Y}|\right). \tag{7}$$

### 2.2.1 Minimum Supervision Ratio

As discussed earlier, generalization of SSL frameworks generally requires a *compatibility* assumption on the hypothesis set $\mathcal{F}$ and data distribution $P_0$. In Appendix C (and in particular, Definition C.4), we introduce a new compatibility function, denoted by MSR, which has the following form: $\mathrm{MSR}_{(\mathcal{F}, P_0)}(\lambda, \mathrm{margin}) : \mathbb{R} \cup \{\pm\infty\} \times \mathbb{R}_{\geq 0} \to [0, 1]$. Intuitively, $\mathrm{MSR}_{(\mathcal{F}, P_0)}$ quantifies the strength of information theoretic relation between the marginal measure $P_{0\boldsymbol{X}}$ and the conditional $P_{0|\boldsymbol{X}}$. It also measures the richness of $\mathcal{F}$ to learn such relations. Due to Theorem 3, in order to bound the true risk when unlabeled data are involved, one needs $\eta \geq \mathrm{MSR}_{(\mathcal{F}, P_0)}(\lambda, \mathrm{margin})$, for some $\lambda$ and $\mathrm{margin} \geq 0$. Here, $\lambda$ denotes the pessimism of the learner and $\mathrm{margin} \geq 0$ specifies a safety margin for small-size datasets. MSR is an increasing function w.r.t. $\mathrm{margin}$, while it decreases with $\lambda$. In particular, $\mathrm{MSR}_{(\mathcal{F}, P_0)}(+\infty, \mathrm{margin}) = 0$, for all $\mathrm{margin} \geq 0$.

For a negative $\lambda$ (optimistic learning), MSR remains small as long as there exists a strong dependency between $P_{0\boldsymbol{X}}$ and label conditionals $P_{0|\boldsymbol{X}}$. This dependency can be obtained, for example, by the *cluster assumption*. Additionally, some loss functions in $\mathcal{F}$ need to be capable of capturing such dependency, e.g. by resembling the true negative log-likelihood $-\log P_0(\boldsymbol{X}, y)$. Conversely, absence of such properties will increase the MSR toward 1, which forces the learner to choose a large $\lambda$ (in the extreme case $+\infty$) in order to use the bound of Theorem 3. Not to mention that a large $\lambda$ increases the empirical loss and loosens the bound. This fact, however, should not be surprising since improper usage of unlabeled data in certain cases can degrade the generalization. Lemma C.3 (supplementary) shows that one can analytically compute MSR function for a particular case of interest, i.e. when cluster assumption holds for $P_0$ and the loss function family $\mathcal{F}$ is chosen properly.

This way, the following Theorem gives a generalization bound for (3):

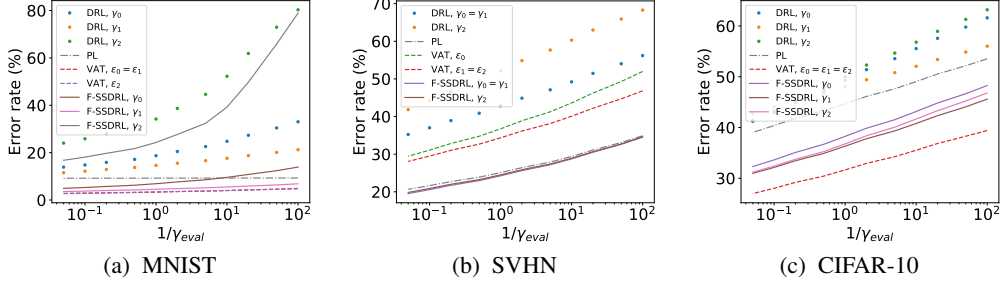

<center>(a) MNIST &emsp;&emsp;&emsp;&emsp; (b) SVHN &emsp;&emsp;&emsp;&emsp; (c) CIFAR-10</center>

Figure 1: Comparison of the test error-rates on adversarial examples attained via [9] among different methods.

**Theorem 3** (Generalization). *For a space $\mathcal{Z} \triangleq \mathcal{X} \times \mathcal{Y}$, assume the set of continuous functions $\mathcal{L} \triangleq \{\ell(\cdot;\theta) \mid \theta \in \Theta\}$, with $\ell(\cdot;\theta) : \mathcal{Z} \to \mathbb{R}$ and $\|\ell\|_\infty \leq B$ for some $B \geq 0$. For $\gamma \geq 0$, $\mathbf{z} \in \mathcal{Z}$ and $\theta \in \Theta$ let $\phi_\gamma(\mathbf{z};\theta)$ to be defined as in (4), and $\Phi \triangleq \{\phi_\gamma(\cdot;\theta) \mid \theta \in \Theta\}$. For a supervision ratio $\eta \in [0,1]$, assume a partially labeled dataset $\mathbf{D} = \{(\mathbf{X}_i, y_i)\}_{i=1}^n$ including $n$ i.i.d. samples drawn from $P_0 \in M(\mathcal{Z})$, where labels can be observed with probability of $\eta$, independently. For $0 < \delta \leq 1$ and $\lambda \in \mathbb{R} \cup \{\pm\infty\}$, assume $\eta$ satisfies the following condition:*

$$\eta \geq \mathrm{MSR}_{(\Phi, P_0)}\left(\lambda, 4B\sqrt{\frac{\log(1/\delta)}{2n}} + 4\mathcal{R}_{n,(\epsilon,\eta)}^{(\mathrm{SSM})}(\mathcal{L})\right). \tag{8}$$

*Then, with probability at least $1 - \delta$, the following bound holds for all $\epsilon \geq 0$:*

$$\sup_{P \in \mathcal{B}_\epsilon(P_0)} \mathbb{E}_P\{\ell(\mathbf{Z};\theta^*)\} \leq \min_{\theta \in \Theta}\hat{R}_{\mathrm{SSAR}}(\theta;\mathbf{D}) + 2B\sqrt{\frac{\log(1/\delta)}{2n}} + 2\mathcal{R}_{n,(\epsilon,\eta)}^{(\mathrm{SSM})}(\mathcal{L}), \tag{9}$$

*where $\theta^*$ is the minimizer of $\hat{R}_{\mathrm{SSAR}}(\theta;\mathbf{D})$.*

Proof of Theorem 3 is given in Appendix D. Condition in (8) can always be satisfied based on Lemma C.2, as long as $\lambda$ and $n$ are sufficiently large and $\mathcal{L}$ is adversarially learnable. A strongly-compatible pair $(\Phi, P_0)$ encourages optimism, where learner can choose a negative $\lambda$. However, in some situations increasing $\lambda$ might be necessary for (8) to hold; In fact, for a weakly-compatible $(\Phi, P_0)$, $\lambda$ must be positive or even $+\infty$ (the latter always satisfies (8) regardless of $n$ or $\eta$). Note that a larger $\lambda$ increases the empirical risk $\hat{R}_{\mathrm{SSAR}}(\theta^*;\mathbf{D})$, which also increases the bound in (9). Interestingly, $\lambda = +\infty$ coincides with the setting of [32], which makes it as a special case of our analysis. The limiting cases of Theorem 3, i.e. $\epsilon = 0$ and $\eta = 1$, lead to a new bound for non-robust SSL, and an existing bound for the supervised DRL of [9], respectively.

## 3 Experimental Results

This section demonstrates our experimental results on some real-world datasets, and also compares SSDRL with its state-of-the-art rival methodologies. Deep Neural Networks (DNN) are considered for the loss $\{\ell(\cdot;\theta) \mid \theta \in \Theta\}$. Architecture and other specifications about our DNNs are explained in details in Appendix A. The rival frameworks in this section are Virtual Adversarial Training (VAT) [4], Pseudo-Labeling (PL) [19], and the supervised DRL of [9], which we simply denote as DRL. We have also implemented a fast version of SSDRL, called F-SSDRL, where for each unlabeled training sample considers only a limited number of *more favorable* labels in Algorithm 1. Here, by *more favorable* labels, we refer to those labels that result in a smaller non-robust loss $\ell(\cdot;\theta)$. As a result, F-SSDRL runs much faster than SSDRL without much degradation in performance. Surprisingly, we found out that F-SSDRL often yields even better performances in practice compared to SSDRL (see Appendix A for more details).

Figure 1 shows the misclassification rate vs. $\gamma^{-1}$ on adversarial test examples attained by computing $\phi_\gamma(\cdot;\theta)$ (same attack strategy as [9]). Recall $\gamma$ as the dual-counterpart of the Wasserstein radius $\epsilon$ in (2). Thus, $\gamma^{-1}$ somehow quantifies the strength of the adversarial attacks, as suggested by [9]. Results

<center>7</center>

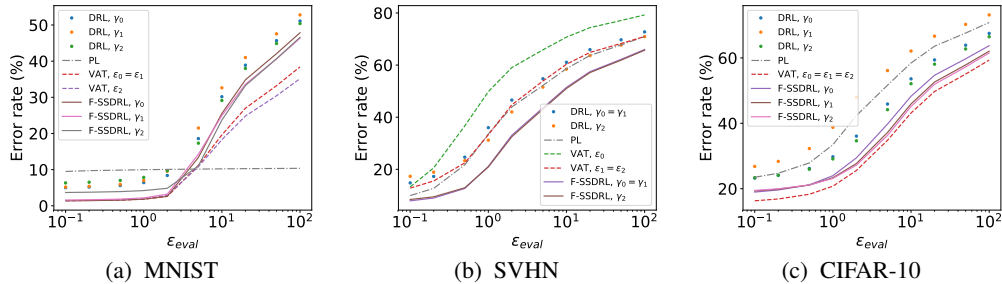

(a) MNIST        (b) SVHN        (c) CIFAR-10

Figure 2: Comparison of the test error-rates on adversarial examples calculated by PGM [38], under $\ell_2$-norm constraint.

| Method | Test Error-Rate(%) | | |
|---|---|---|---|
| | MNIST | SVHN | CIFAR-10 |
| DRL | | | |
| $\gamma_1$ | 4.67±0.38 | 10.89±0.53 | 21.62±0.40 |
| $\gamma_2$ | 4.77±0.15 | 10.89±0.53 | 23.77±0.65 |
| $\gamma_3$ | 5.95±0.13 | 14.45±0.93 | 21.97±0.35 |
| PL | 8.70±0.47 | 6.39±0.46 | 21.19±0.25 |
| VAT | | | |
| $\varepsilon_1$ | 1.30±0.04 | 5.47±0.33 | 15.19±0.55 |
| $\varepsilon_2$ | 1.30±0.04 | 7.01±0.24 | 15.19±0.55 |
| $\varepsilon_3$ | 1.32±0.10 | 7.01±0.24 | 15.19±0.55 |
| F-SSDRL | | | |
| $\gamma_1$ | 1.29±0.09 | 6.19±0.22 | 17.94±0.20 |
| $\gamma_2$ | 1.51±0.03 | 6.19±0.22 | 18.35±0.24 |
| $\gamma_3$ | 3.58±1.13 | 6.74±0.22 | 18.64±0.16 |

Table 2: Test error-rates on clean examples. For DRL, VAT and F-SSDRL, rows 1 to 3 correspond to the parameter ($\gamma_i$ for DRL and F-SSDRL, and $\varepsilon_i$ for VAT) that yields the lowest error rates on: ($i = 1$) clean examples, ($i = 2$) adversarial examples by [9], and ($i = 3$) adversarial examples by PGM, respectively.

have been depicted for MNIST, SVHN and CIFAR-10 datasets. Figure 2 demonstrates the same procedure for adversarial examples generated by Projected-Gradient Method (PGM) [38]; In this case, the error-rate is depicted vs. PGM's *strength of attack*, i.e. $\varepsilon$. For VAT and SSDRL, curves have been shown for different choices of hyper-parameters, i.e., $(\gamma_i \text{ or } \varepsilon_i)$, $i = 1, 2, 3$, which correspond to the lowest error rates on: ($i = 1$) clean examples, ($i = 2$) adversarial examples by [9], and ($i = 3$) adversarial examples by PGM, respectively. Values of $(\gamma_i, \varepsilon_i)$, and the choices of $\lambda$, transportation cost $c$, and the supervision ratio $\eta$ with more details on the experiments can be found in Appendix A.

According to Figures 1 and 2, the proposed method is always superior to DRL and PL. Also, SSDRL outperforms VAT on SVHN dataset regardless of the attack type, while it has a comparable error-rate on MNIST and CIFAR-10 based on Figures 1a and 2c, respectively. The superiority over DRL highlights the fact that exploitation of unlabeled data has improved the performance. However, SSDRL under-performs VAT on MNIST and CIFAR-10 datasets if the order of attacks are reversed, even though performances are still close. According to Figure 2a, accuracy of PL degrades quite slowly as PGM's $\varepsilon$ increases, although the loss values increase in Figure A.7a. This phenomenon is due to the fact that the adversarial directions for increasing the *loss* and *error-rate* are not correlated in this particular case.

Table 2 shows the test error-rates on clean examples for F-SSDRL, VAT, PL and DRL on MNIST, SVHN and CIFAR-10 datasets. In fact, Table 2 quantifies the non-adversarial generalization that one can attain in practice via distributional robustness. Again, F-SSDRL outperforms both PL and DRL in all experimental settings. It also surpasses VAT on SVHN dataset. F-SSDRL under-performs VAT on MNIST and CIFAR-10, however, the differences between error-rates remain small which means the two methods have comparable performances.

## 4 Conclusions

This paper investigates the application of distributionally robust learning in partially labeled datasets. The core idea is to take a well-known semi-supervised framework, known as self-learning, and make it robust to adversarial attacks. A novel framework, called SSDRL, has been proposed which encompasses many existing methods such as Pseud-Labeling (PL) and EM algorithm as its special cases. Computational complexity of our method is shown to be comparable with its supervised counterparts. We have derived convergence and generalization guarantees for SSDRL, where for the latter, a number of novel complexity measures are proposed. In particular, an adversarial extension of Rademacher complexity is proposed and shown to converge to zero for almost all practical learning frameworks, including neural networks, that have a finite VC-dimension. Moreover, our theoretical analysis reveals a more general and fundamental condition to assess the role of unlabeled data in generalization by introducing a new complexity measure called Minimum Supervision Ratio (MSR). This is in contrast to many existing works that need more restrictive assumptions, such as *cluster assumption* to be applicable. Computer simulation on real-world datasets demonstrate a comparable-to-superior performance for SSDRL compared with those of the state-of-the-art. In future, we try to improve the generalization, for example, by empirically computing the MSR. Also, fitting more SSL methods into the core idea of our work is another research direction.

## Footnotes

[1]In scenarios where $|\mathcal{Y}|$ is very large, one can employ heuristic methods to reduce the set of *possible labels* for an unlabeled data sample and gain more efficiency at the expense of degradation in performance

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
