[Supplementary Material · NeurIPS_supp.pdf]

# Robustness to Adversarial Perturbations in Learning from Incomplete Data
# (Supplementary Document)

**Amir Najafi**
Department of Computer Engineering
Sharif University of Technology
Tehran, Iran
najafy@ce.sharif.edu

**Shin-ichi Maeda**
Preferred Networks, Inc.
Tokyo, Japan
ichi@preferred.jp

**Masanori Koyama**
Preferred Networks, Inc.
Tokyo, Japan
masomatics@preferred.jp

**Takeru Miyato**
Preferred Networks, Inc.
Tokyo, Japan
miyato@preferred.jp

## A  Additional Simulations and Experimental Settings

This section presents a number of additional experiments w.r.t. the proposed method and shows more comparison with rival methodologies. We also give an extensive description of the experimental setting that we have used for our computer simulations.

### A.1  Additional Simulations

Figure A.1 depicts the error-rate corresponding to DRL, SSDRL and F-SSDRL as a function of $\gamma^{-1}$, on adversarial examples in the MNIST dataset which are generated via the maximization problem $\text{argmax}_{\boldsymbol{z}'} \ \ell\left(\boldsymbol{z}';\theta\right) - \gamma c\left(\boldsymbol{z}';\cdot\right)$ (as described in [1]). Unlike Figures 1 and 2, we have shown the results for a range of values of $\gamma$ and $\lambda$, in order to experimentally measure the sensitivity of our method to these hyper-parameters. Also, we have performed the same procedure for DRL for the sake of comparison. In particular, Figure A.1a shows the comparison between DRL and SSDRL (with $\lambda$ set to $-1$ for SSDRL) and different values of $\gamma$. As it is evident for the majority of cases ($\gamma \geq 0.05$), SSDRL performs much better than DRL. This result indicates that employing the unlabeled data samples improves the generalization, which is highly favorable. Figure A.1b depicts the comparison between F-SSDRL and the original SSDRL (again $\lambda$ is set to -1 for SSDRL). Figure A.1c shows the effect of varying $\lambda$ (with $\gamma$ fixed to 1). Surprisingly, the error-rate experiences a drastic jump when one changes the sign of $\lambda$, which indicates a trade-off between *optimism* and *pessimism*. This result might be related to the fact that for the case of MNIST dataset, learned neural networks on the labeled part of the dataset are sufficiently reliable, and thus encourage the user to employ an optimistic approach (i.e., setting a negative $\lambda$) in order to improve the performance. However, while the sign of $\lambda$ is fixed, error-rate does not show that much sensitivity to the magnitude of $\lambda$, which can be noted as a point of strength for SSDRL.

Figure A.2 is a complete version of Figure 1 from Section 3, where the performances of SSDRL, fully-supervised DRL, PL and VAT are extensively investigated on three benchmark datasets, i.e. MNIST, SVHN and CIFAR-10. SSDRL and VAT have been tested with a variety of their corresponding hyper-parameters $\gamma$ and $\epsilon$. Figure A.3 is the counterpart of Figure A.2, where the *attack* strategy is replaced with Projected-Gradient Method (PGM). Again, error-rates have been depicted as a function of PGM's *attack strength*, i.e. $\varepsilon$. Even though more variation in hyper-parameters has been considered,

Figure A.1: Error rates on adversarial examples generated via the algorithm in [1] vs. $\gamma_{\text{eval}}^{-1}$ on the MNIST dataset.

(a) MNIST      (b) SVHN      (c) CIFAR10

Figure A.2: Comparison of test error rates of SSDRL, DRL, PL and VAT on the adversarial examples generated via [1] on different datasets. $\lambda$ is set to $-1$.

we have not observed any significant sensitivity that is caused by a slight change of parameter values. As a result, one can say that DRL, SSDRL and VAT are all stable algorithms w.r.t. to their parameter values, at least up to some certain levels.

Figures A.4 and A.5 represent the performance (again in terms of error-rate) over clean examples from different datasets, and for SSDRL and VAT, respectively. In Figure A.4, different values of $\gamma$ have been used for training and the test error-rate is depicted as a function of $\gamma^{-1}$. Also, $\lambda$ is set to $-1$ for SSDRL. Apparently, SSDRL (or F-SSDRL), for a particular range of parameters, over-fits during the training stage on MNIST and as a result its performance is degraded when compared to that of DRL. However, SSDRL outperforms DRL (its fully-supervised counterpart) on SVHN and CIFAR-10 datasets. Also, SSDRL and VAT have comparable performances on clean examples, specifically on SVHN and CIFAR-10 datasets. This observation is in agreement with Table 2.

So far, the performance of SSDRL has been demonstrated w.r.t. its misclassification rate. We have also provided extensive experimental results on the value of adversarial loss $\phi_\gamma$, which are crucial for the computation of our generalization bound in Section 2.2. Figure A.6 shows the average adversarial loss, i.e. $\frac{1}{n_{\text{test}}} \sum_{i \in \text{test}} \phi_\gamma(\mathbf{z}_i)$, for different methods and on different datasets. $\lambda$ is set to $-1$ for SSDRL. Again, it should be noted that the adversarial examples used in Figures A.2 and A.6 are generated via the procedure described in [1]. Figure A.7 is the counterpart of Figure A.6, where the attack strategy is replaced with Projected-Gradient Method (PGM). As a result, adversarial loss values

(a) MNIST       (b) SVHN       (c) CIFAR10

Figure A.3: Comparison of the test error rates on adversarial examples computed by Projected-Gradient Method (PGM) [2] under $\ell_2$ norm constraint.

(a) MNIST       (b) SVHN       (c) CIFAR10

Figure A.4: Test error rates of distributionally robust learning methods on clean examples. The solid lines and shaded regions around them represent the mean and standard deviation of results over multiple random seeds, respectively.

have been depicted as a function of PGM's strength of attack, i.e. $\varepsilon$. As can be seen, SSDRL (or its fast version F-SSDRL) are always among the few methods that generate the smallest adversarial loss values, regardless of the strength of attacks. This means that the proposed method can establish a reliable certificate of robustness for test samples via Theorem 3. Note that VAT, another method that performs well in practice in terms of error-rate, does not have any theoretical guarantees.

## A.2 Experimental Settings

In this part, we present a detailed description of the experimental settings which have been used for Section 3. It should be noted that the majority of the settings used for SVHN and CIFAR-10 datasets follow the same procedure as described in [3].

### A.2.1 Real-world Datasets

Three main datasets have been used during the experiments: MNIST, SVHN and CIFAR-10.

- The MNIST dataset consists of $28 \times 28$ pixel, gray-scale images of handwritten digits together with their corresponding labels. Each label is a natural number from 0 to 9. The number of training examples and test examples in the dataset are $60,000$ and $10,000$, respectively.

- The SVHN dataset consists of $32 \times 32 \times 3$ pixel RGB images of street view house numbers with their corresponding labels. Again, labels are natural numbers ranging from 0 to 9. The number of training and test samples in the dataset are $73,257$ and $26,032$, respectively.

(a) MNIST                (b) SVHN                (c) CIFAR10

Figure A.5: Test error rates of VAT on clean examples with different $\epsilon$. The solid lines and shaded regions around them represent the mean and standard deviation of results over multiple random seeds, respectively.

(a) MNIST                (b) SVHN                (c) CIFAR-10

Figure A.6: Comparison of the average adversarial loss among different methods.

- CIFAR-10 dataset consists of $32 \times 32 \times 3$ pixel RGB images of categorized objects, i.e., cars, trucks, planes, animals, and humans. The number of training examples and test examples in the dataset are $50,000$ and $10,000$, respectively. For CIFAR-10 dataset, we conducted Zero-phase Component Analysis (ZCA) as a pre-processing stage prior to the experiments.

### A.2.2 Supervision Ratio and Training Data-points

In order to create a dataset (training+testing) for the semi-supervised learning task in the paper, we selected a subset of size $1,000$ as the labeled dataset from MNIST and SVHN, while the size goes up to $4,000$ for CIFAR-10. The rest of the samples in the training partition are treated as unlabeled data. We repeated the experiment three times with different choices of labeled and unlabeled data-points on all of the three datasets. For MNIST, a mini-batch of size $64$ is used for both the labeled and unlabeled term, and for SVHN and CIFAR-10, a mini-batch of size $32$ is used for the calculation of the labeled term, while a mini-batch of size $128$ is employed for the unlabeled term during the implementation of each method. We trained each model with $50,000$ updates for MNIST and $48,000$ updates for SVHN and CIFAR10. We have used ADAM optimizer in the training stage. In this regard, the initial learning rate of ADAM is set to $0.001$ and then linearly decayed over the last $10,000$ updates for MNIST, and the last $16,000$ updates for SVHN and CIFAR-10.

As for the transportation cost function $c$, we follow the work presented in [1] and thus employed the following cost function throughout all our experiments:

$$c(\boldsymbol{z}, \boldsymbol{z}') = \|\boldsymbol{z} - \boldsymbol{z}'\|_2^2 + \infty \cdot \mathbf{1}\{y \neq y'\}, \tag{A.1}$$

where $\mathbf{1}(\cdot)$ is an indicator function which returns 1 if its input condition holds and zero, otherwise. It should be noted that this choice is solely for the sake of simplicity, and as described before, every valid lower semi-continuous function is a legitimate choice for $c$.

| (a) MNIST | (b) SVHN | (c) CIFAR-10 |

Figure A.7: Comparison of the loss on adversarial examples calculated by projected-gradient method (PGM) [2] under $\ell_2$ norm constraint.

Also, the *pessimism/optimism trade-off* parameter $\lambda$ is always set to $-1$, except when stated otherwise. This option yields certain degrees of optimism during the learning stage, which is motivated by the fact that Deep Neural Networks (DNN) have already proven to work well on all the above-mentioned three datasets. Thus, trusting the learner to assign soft pseudo-labels to the unlabeled data is somehow encouraged which in turn indicates a negative value for $\lambda$.

### A.2.3 Creating Adversarial Examples

To solve the inner maximization problem in (4) and (E.2) for each pair of $(\boldsymbol{X}, y) \in \mathcal{X} \times \mathcal{Y}$, we simply apply *Gradient Ascent* with the following update rule:

$$\boldsymbol{X}_{t+1} = \boldsymbol{X}_t + r_t \nabla_{\boldsymbol{X}_t} \left[ \ell((\boldsymbol{X}_t, y); \theta) - \gamma c\left((\boldsymbol{X}_t, y), (\boldsymbol{X}, y)\right) \right], \tag{A.2}$$

where the initial value $\boldsymbol{X}_0$ is set to $\boldsymbol{X}$, and the ascent rate is defined as $r_t \triangleq \frac{\kappa/\gamma}{(t+1)}$, where $\kappa$ is a hyper-parameter. We set $\kappa$ to 1.0 for MNIST and CIFAR-10, and 0.5 for SVHN. During the training, we repeat the update in (A.2) 5 times for both the DRL and SSDRL method. However, we repeat it 15 times during the evaluation.

While generating the adversarial examples via the Projected-Gradient Method (PGM), we applied the following update rule which is also used in some previous works in this area [1, 2]:

$$\boldsymbol{X}_{t+1} = \mathrm{Proj}_{\boldsymbol{X}, \epsilon} \left( \boldsymbol{X}_t + \xi \overline{\nabla_{\boldsymbol{X}_t} \ell((\boldsymbol{X}_t, y); \theta)} \right), \tag{A.3}$$

where $\mathrm{Proj}_{\boldsymbol{X}, \epsilon}$ represents the projection operator to an $\epsilon$-ball (w.r.t. $\ell_2$ norm) centered on $\boldsymbol{X}$. Also, $\bar{\boldsymbol{v}}$ for an arbitrary vector $\boldsymbol{v}$ denotes its normalized version, which is mathematically defined as $\boldsymbol{v}/\|\boldsymbol{v}\|_2$ under the $\ell_2$-norm constraint. We have defined the length parameter $\xi$ as $\epsilon/\log(T)$, where $T$ denotes the number of iterations of the update (A.3). Accordingly, we set $T = 15$.

### A.2.4 Architecture of Deep Neural Networks

A class of Convolutional Neural Networks (CNN) has been used for the loss function set $\mathcal{L} = \{\ell\left(\cdot; \theta\right), \theta \in \Theta\}$. Table 1 shows the CNN models used in our experiments. We use ELU [4] for the activation function in MNIST, and leakyReLU (lReLU) [5] for SVHN and CIFAR-10. In the CNNs used for SVHN and CIFAR-10, all the convolutional layers as well as the fully connected (or equivalently dense) layers are followed by batch normalization [6], except for the fully connected layer on CIFAR-10. The slopes of all lReLU in the network are set to 0.1.

## B Additional Definitions

Additional definitions and/or notations are presented in this section.

| (a) For MNIST | (b) For SVHN and CIFAR-10 |
| --- | --- |
| $28{\times}28$ gray-scale image | $32{\times}32$ RGB image |
| $4{\times}4$ conv. stride 2, 64 ELU<br>$4{\times}4$ conv. stride 2, 64 ELU<br>$4{\times}4$ conv. stride 2, 64 ELU | $3{\times}3$ conv. 128 lReLU<br>$3{\times}3$ conv. 128 lReLU<br>$3{\times}3$ conv. 128 lReLU |
| global average pool | $2{\times}2$ max-pool, stride 2<br>dropout, $p = 0.5$ |
| dense $64 \to 10$ | $3{\times}3$ conv. 256 lReLU<br>$3{\times}3$ conv. 256 lReLU<br>$3{\times}3$ conv. 256 lReLU |
| 10-way softmax | $2{\times}2$ max-pool, stride 2<br>dropout, $p = 0.5$ |
|  | $3{\times}3$ conv. 512 lReLU<br>$1{\times}1$ conv. 256 lReLU<br>$1{\times}1$ conv. 128 lReLU |
|  | global average pool |
|  | dense $128 \to 10$ |
|  | 10-way softmax |

Table 1: CNN models used in our experiments. The deep structures that have been used for SVHN and CIFAR-10 datasets are different from the one used for MNIST. The specifications that correspond to each structure are inspired from [3].

**Definition B.1** (Wasserstein distance). *Assume $c : \mathcal{Z} \times \mathcal{Z} \to [0, +\infty)$ to be a non-negative and lower semi-continuous function, where $c\left(\boldsymbol{z}, \boldsymbol{z}\right) = 0$ for all $\boldsymbol{z} \in \mathcal{Z}$. Then, the Wasserstein distance between two distributions $P$ and $Q$ in $M\left(\mathcal{Z}\right)$ with respect to cost $c$ is defined as:*

$$W_c\left(P, Q\right) \triangleq \inf_{\mu \in M(\mathcal{Z}^2)} \int c\left(\boldsymbol{z}, \boldsymbol{z}'\right) \mathrm{d}\mu\left(\boldsymbol{z}, \boldsymbol{z}'\right) \tag{B.1}$$
$$\text{subject to} \quad \mu\left(\cdot, \mathcal{Z}\right) = P\,, \ \mu\left(\mathcal{Z}, \cdot\right) = Q,$$

*where $M\left(\mathcal{Z}^2\right)$ represents the set of all couplings between any two random variables supported on $\mathcal{Z}$. Also, $\mu\left(\mathcal{Z}, \cdot\right)$ and $\mu\left(\cdot, \mathcal{Z}\right)$ denote the marginals of $\mu$ w.r.t. the first and second variables, respectively.*

## C   Minimum Supervision Ratio: Definition and Implications

In this section, we present some complementary discussions with respect to our generalization bound in Section 2.2. In particular, the mathematical definition and intuitive implications behind one of our proposed complexity measures, i.e. the Minimum Supervision Ratio, are explained in details.

In order to better understand the intuition behind the proposed optimization programs in (2) or (3), it is necessary to investigate them under the asymptotic regime of $n \to \infty$. In this regard, this section provides a rigorous mathematical framework to study the semi-supervised learning in general (and its distributionally robust extension in particular), under the specific problem setting of this paper. We then provide conditions on the hypothesis set and data-generating distribution, under which unlabeled data can help the overall learning procedure. Final bounds on the performance improvement through incorporation of unlabeled samples (which is mostly from the generalization aspect), are given with mathematical details in Theorem 3 and its proof. In order to achieve the above-mentioned goal, first let us make the following definition:

**Definition C.1.** *For a feature space $\mathcal{X}$ and a finite label set $\mathcal{Y}$, the conditional composition of a distribution $P \in M\left(\mathcal{X} \times \mathcal{Y}\right)$ with a conditional distribution $\Omega \in M^{\mathcal{X}}\left(\mathcal{Y}\right)$ through a supervision ratio of $0 \leq \eta \leq 1$, denoted by $\mathrm{comp}\left(P, \Omega, \eta\right) \in M\left(\mathcal{X} \times \mathcal{Y}\right)$, is defined as*

$$\mathrm{comp}\left(P, \Omega, \eta\right)\left(\boldsymbol{X}, y\right) \triangleq \eta P\left(\boldsymbol{X}, y\right) + \left(1 - \eta\right) \Omega\left(y | \boldsymbol{X}\right) \left(\sum_{y' \in \mathcal{Y}} P\left(\boldsymbol{X}, y'\right)\right). \tag{C.1}$$

It can be easily verified that the following properties hold for the conditional composition distribution of any two corresponding distributions:

$$\text{comp}\,(P,\Omega,\eta)_{\boldsymbol{X}} = P_{\boldsymbol{X}} \quad , \quad \text{comp}\,(P,\Omega,\eta)_{|\boldsymbol{X}} = \eta P_{|\boldsymbol{X}} + (1-\eta)\,\Omega_{|\boldsymbol{X}}, \tag{C.2}$$

where the first relation means: the marginal of the composition distribution w.r.t. $\boldsymbol{X}$ (which is a measure supported on $\mathcal{X}$) is the same as that of $P$, while the second property states that: conditional distribution over $\mathcal{Y}$ (given $\boldsymbol{X} \in \mathcal{X}$) is a weighted mixture of conditional distributions $P_{|\boldsymbol{X}}$ and $\Omega_{|\boldsymbol{X}}$.

An interesting asymptotic property of a consistent distribution set (see Definition 1) is that, given both fully and partially-observed samples in $\boldsymbol{D}$ are i.i.d. samples generated from a single arbitrary distribution $P_0 \in M\,(\mathcal{X} \times \mathcal{Y})$, the following relation holds almost surely w.r.t. $P_0$:

$$\lim_{n\to\infty} \hat{\mathcal{P}}\,(\boldsymbol{D}) \overset{a.s.}{=} \left\{ \text{comp}\left(P_0,\Omega,\eta = \lim_{n\to\infty}\frac{n_1}{n}\right) \middle| \Omega \in M^{\mathcal{X}}\,(\mathcal{Y}) \right\}, \tag{C.3}$$

where the asymptotic equality in the above relation corresponds to a member-wise convergence between the two sets. Consequently, rewriting (3) in the asymptotic regime of $n \to \infty$ would give us the following equalities:

$$\lim_{n\to\infty} \hat{R}_{\text{SSAR}}\,(\theta;\boldsymbol{D}) \overset{a.s.}{=} \mathbb{E}_{P_0} \left\{ \hat{R}_{\text{SSAR}}\,(\theta;\boldsymbol{D}) \right\} \tag{C.4}$$

$$= \eta \mathbb{E}_{(\boldsymbol{X},y)\sim P_0} \left\{ \phi_\gamma\,(\boldsymbol{X},y;\theta) \right\} + (1-\eta)\,\mathbb{E}_{\boldsymbol{X}\sim P_{0\boldsymbol{X}}} \left\{ \underset{y\in\mathcal{Y}}{\overset{(\lambda)}{\text{softmin}}} \left\{ \phi_\gamma\,(\boldsymbol{X},y;\theta) \right\} \right\}.$$

The first term in the r.h.s. of (C.4) is proportional to the true risk which we intend to bound. However, the second term models the asymptotic effect of unlabeled data for a fixed supervision ratio $\eta$. The main question that we try to answer in this section can be intuitively stated as: under what conditions, the second term becomes *approximately proportional* to the true risk as well?

Before investigating the above question in more theoretical details, a closer look at the semi-supervised adversarial risk $\hat{R}_{\text{SSAR}}$ reveals that

$$\frac{\partial}{\partial\lambda} \hat{R}_{\text{SSAR}}\,(\theta;\boldsymbol{D}) \geq 0. \tag{C.5}$$

This fact implies that by decreasing $\lambda$, one can also decrease $\hat{R}_{\text{SSAR}}$ (at least in the majority of non-trivial scenarios). This issue has been previously mentioned in Section 2, which indicates that *optimism* always results in lower empirical risks. But how does this strategy affect the true expected loss, i.e. $\mathbb{E}_{P_0}\{\phi_\gamma\,(\boldsymbol{Z};\theta)\}$? On the other hand, moving $\lambda$ toward $+\infty$ guarantees that the learner is minimizing a legitimate upper-bound of the true risk, i.e. extreme pessimism, however, this also increases the empirical risk. Again, one could ask is it really necessary to be so pessimistic?

In order to answer the above questions, we introduce a new compatibility measure function for a function set $\Phi \subset \mathbb{R}^{\mathcal{X}\times\mathcal{Y}}$ and distribution $P_0$, denoted by *minimal supervision ratio* or $\text{MSR}_{(\Phi,P_0)}$ : $\mathbb{R} \times \mathbb{R}_{\geq 0} \to [0,1]$. We then show that as long as a particular inequality holds among parameters such as $n$, $\lambda$ and $\eta$ according to $\text{MSR}_{(\Phi,P_0)}$, one can guarantee minimizing a valid upper-bound for the true risk, while avoiding the extreme pessimism of [7] (less harm to the empirical risk minimization). In order to do so, first let us introduce a number of useful additional tools:

**Definition C.2.** *Assume function class $\Phi \subseteq \mathbb{R}^{(\mathcal{X}\times\mathcal{Y})}$ and distribution $P_0 \in M\,(\mathcal{X} \times \mathcal{Y})$ for a finite label-set $\mathcal{Y}$. For the ease of notation, let $\phi_{\boldsymbol{X}} \triangleq \phi\,(\boldsymbol{X},\cdot) \in \mathbb{R}^{\mathcal{Y}}$ for $\forall \boldsymbol{X} \in \mathcal{X}$. Then, $\rho_\lambda\,(\phi)$ for $\phi \in \Phi$ and $\lambda \in \mathbb{R} \cup \{\pm\infty\}$ is defined as*

$$\rho_\lambda\,(\phi) \triangleq \mathbb{E}_{P_{0\boldsymbol{X}}} \left\{ \underset{y\in\mathcal{Y}}{\overset{(\lambda)}{\text{softmin}}} \left\{ \phi_{\boldsymbol{X}} \right\} \right\} - \mathbb{E}_{P_0}\,\{\phi\}. \tag{C.6}$$

As it becomes evident in the proceeding arguments of this section, the introduced functional in Definition C.2, i.e. $\rho_\lambda$, plays an important role in determining the relation of expected (or asymptotic) semi-supervised risk with the true (supervised) one. Mathematically speaking, enforcing $\rho_\lambda\,(\phi)$ for $\phi = \phi_\gamma\,(\cdot;\theta)$ to remain non-negative guarantees that $\mathbb{E}_{\boldsymbol{D}\sim P_0} \left\{ \hat{R}_{\text{SSAR}}\,(\theta;\boldsymbol{D}) \right\} \geq \mathbb{E}_{P_0}\{\phi_\gamma\,(\cdot;\theta)\}$ for any $\theta \in \Theta$. This allows us to upper-bound the true risk with the value of $\hat{R}_{\text{SSAR}}$ computed for that particular $\theta$. Surprisingly, this condition can always be satisfied by choosing $\lambda = +\infty$ (extreme pessimism). This configuration, in the special non-robust case, coincides with the framework presented in [7].

**Lemma C.1.** *For any function set $\Phi \subseteq \mathbb{R}^{(\mathcal{X} \times \mathcal{Y})}$ and distribution $P_0 \in M(\mathcal{X} \times \mathcal{Y})$, we have $\rho_\infty(\phi) \geq 0$ for all $\phi \in \Phi$.*

*Proof.* $P_{0|\boldsymbol{x}}$ is a distribution over $\mathcal{Y}$, thus can be considered as a vector in a simplex, i.e. all components are non-negative and sum up to one. Then, the lemma's argument can be justified by the fact that

$$\langle \phi_{\boldsymbol{X}} | P_{0|\boldsymbol{x}} \rangle \leq \max_{y \in \mathcal{Y}} \phi_{\boldsymbol{X}}, \quad \text{while} \quad \overset{(\infty)}{\underset{y \in \mathcal{Y}}{\text{softmin}}} \{\phi_{\boldsymbol{X}}\} = \max_{y \in \mathcal{Y}} \phi_{\boldsymbol{X}}, \tag{C.7}$$

where $\langle \cdot | \cdot \rangle$ denotes the inner product. More precisely, one can write:

$$\rho_\infty(\phi) = \mathbb{E}_{P_{0\boldsymbol{X}}} \left\{ \overset{(\infty)}{\underset{y \in \mathcal{Y}}{\text{softmin}}} \{\phi_{\boldsymbol{X}}\} \right\} - \mathbb{E}_{P_0} \{\phi\}$$

$$= \mathbb{E}_{P_{0\boldsymbol{X}}} \left\{ \overset{(\infty)}{\underset{y \in \mathcal{Y}}{\text{softmin}}} \{\phi_{\boldsymbol{X}}\} - \langle \phi_{\boldsymbol{X}} | P_{0|\boldsymbol{x}} \rangle \right\} \geq 0.$$

The last inequality is a direct result of the fact that inside of the expectation operator is non-negative. This completes the proof. $\square$

However, we are more interested in those cases where $\lambda$ can be bounded, or even negative, while $\rho_\lambda$ is still non-negative in *some regions* of $\Phi$. The main problem is that the minimizer of (3) (semi-supervised empirical risk) must fall in *those regions*, as well. Otherwise one cannot upper-bound the true risk by minimizing (3). Mathematically speaking, assume $\Phi \triangleq \{\phi_\gamma(\cdot; \theta) : \mathcal{Z} \to \mathbb{R} | \theta \in \Theta\}$ as described in (3). Then, we are interested to see if there exists a non-empty subset of $\Phi$, say $\psi$, such that:

$$\exists \psi \subseteq \Phi \left| \underset{\phi \in \Phi}{\text{argmin}} \ \hat{R}_{\text{SSAR}}(\phi; \boldsymbol{D}) \in \psi \quad \text{and} \quad \rho_\lambda(\phi) \geq 0, \ \forall \phi \in \psi. \tag{C.8}\right.$$

We give a theoretical solution for the non-trivial case of the above-mentioned problem ($\lambda < +\infty$). This way, one can still choose small (or generally negative) values of $\lambda$, which substantially lower the empirical loss and improve the generalization bound. The following definitions provide us with more generalized means to achieve this goal.

**Definition C.3.** *Assume the function set $\Phi \subseteq \mathbb{R}^{(\mathcal{X} \times \mathcal{Y})}$, probability distribution $P_0 \in M(\mathcal{X} \times \mathcal{Y})$, and let us define $\phi^* = \text{argmin}_{\phi \in \Phi} \ \mathbb{E}_{P_0} \{\phi(\boldsymbol{X}, y)\}$. Let $\psi \subseteq \Phi$ to denote a subset of functions in $\Phi$. Then, the loss gap functional $\text{GAP}(\psi)$, and $\Gamma(\psi; \lambda)$ for $\lambda \in \mathbb{R} \cup \{\pm\infty\}$ w.r.t. $P_0$ and $\Phi$ are defined as*

$$\text{GAP}(\psi) \triangleq \inf_{\phi \in \Phi - \psi} \mathbb{E}_{P_0} \{\phi - \phi^*\} \geq 0 \quad, \quad \Gamma(\psi; \lambda) \triangleq \inf_{\phi \in \Phi - \psi} \rho_\lambda(\phi) - \rho_\lambda(\phi^*). \tag{C.9}$$

*For the special case of $\psi = \Phi$, we define $\text{GAP}(\Phi) = \infty$ and $\Gamma(\Phi; \lambda) = 0$, respectively. Also, let us define $\Lambda : 2^\Phi \to \mathbb{R} \cup \{\pm\infty\}$ as*

$$\Lambda(\psi) \triangleq \inf_{\lambda \in \mathbb{R} \cup \{\pm\infty\}} \lambda$$
$$\text{subject to} \quad \rho_\lambda(\phi) \geq 0, \quad \forall \phi \in \psi. \tag{C.10}$$

All the functionals $\text{GAP}$, $\Gamma$ and $\Lambda$ are defined to enable us to capture the properties of a hypothesis set $\Phi$ and a corresponding data distribution $P_0$, inside arbitrary subsets of $\Phi$. Another interesting attribute is that $\text{GAP}$ and $\Lambda$ are not functions of $\lambda$, and correspond to the fundamental features of the pair $(\Phi, P_0)$ in a fully-supervised sense. Note that due to Lemma C.1, $\Lambda(\psi)$ is always well-defined, since its corresponding feasible set cannot be empty. This way, we can present the most important definition in this section, which is the key to provide the generalization bounds derived in Theorem 3 for general semi-supervised learning via self-learning.

**Definition C.4** (Minimum Supervision Ratio). *Assume function set $\Phi \subseteq \mathbb{R}^{(\mathcal{X} \times \mathcal{Y})}$ and distribution $P_0 \in M(\mathcal{X} \times \mathcal{Y})$ for a feature space $\mathcal{X}$ and finite label set $\mathcal{Y}$. Then, the minimum supervision ratio function, $\text{MSR}_{(\Phi, P_0)} : \mathbb{R} \cup \{\pm\infty\} \times \mathbb{R}_{\geq 0} \to [0, 1]$, is defined as*

$$\text{MSR}_{(\Phi, P_0)}(\lambda, \zeta) \triangleq \inf_{\psi \subseteq \Phi | \Lambda(\psi) \leq \lambda} h\left(1 - \frac{\text{GAP}(\psi) - \zeta}{u(-\Gamma(\psi; \lambda))}\right), \tag{C.11}$$

*for* $\lambda \in \mathbb{R} \cup \{\pm\infty\}$ *and* $\zeta \geq 0$, *where* $u : \mathbb{R} \to \mathbb{R}$ *denotes the ramp function, i.e.* $u(x) = x$, $x \geq 0$ *and* $0$ *otherwise, and* $h(\cdot) \triangleq \min\{1, u(\cdot)\}$. *Also, let* $\mathrm{MSR}_{(\Phi, P_0)}(\lambda, \zeta) = 1$, *in case the feasible set* $\Lambda(\psi) \leq \lambda$ *is empty for an input* $\lambda$.

$\mathrm{MSR}_{(\Phi, P_0)}$ is a learning-theoretic attribute of the pair $(\Phi, P_0)$, and also a central ingredient of Theorem 3. It has the following properties: First, $\mathrm{MSR}_{(\Phi, P_0)}(\lambda, \zeta)$ is an increasing function w.r.t. $\zeta$, and decreasing w.r.t. $\lambda$, for all $\Phi$ and $P_0$. Second, for $\Phi$ and $P_0$, there exist $\lambda \in \mathbb{R} \cup \{\pm\infty\}$ and $\zeta \geq 0$ such that $\mathrm{MSR}_{(\Phi, P_0)}(\lambda, \zeta) = 0$ (see Lemma C.2 below).

**Lemma C.2** (Compatibility Guarantee). *For any function set* $\Phi$ *and a corresponding probability distribution* $P_0$, *there exist* $\lambda \in \mathbb{R} \cup \{\pm\infty\}$ *and* $\zeta \geq 0$ *such that* $\mathrm{MSR}_{(\Phi, P_0)}(\lambda, \zeta) = 0$.

*Proof.* By simple mathematical manipulations, it can be easily verified that

$$\mathrm{MSR}_{(\Phi, P_0)}(\lambda, \zeta) = \inf_{\psi \subseteq \Phi \mid \Lambda(\psi) \leq \lambda} h\left(1 - \frac{\mathrm{GAP}(\psi) - \zeta}{u(-\Gamma(\psi; \lambda))}\right) = 0,$$

$$\Rightarrow \quad \exists \lambda \in \mathbb{R} \cup \{\pm\infty\} \, \Bigg| \, \sup_{\psi \subseteq \Phi \mid \Lambda(\psi) \leq \lambda} \mathrm{GAP}(\psi) + \Gamma(\psi; \lambda) \geq \zeta. \tag{C.12}$$

In this regard, in order to prove the lemma one can alternatively try to show that there exists $\zeta \geq 0$, such that

$$\sup_{\lambda \in \mathbb{R} \cup \{\pm\infty\}} \sup_{\psi \subset \Phi \mid \Lambda(\psi) \leq \lambda} \mathrm{GAP}(\psi) + \Gamma(\psi; \lambda) \geq \zeta. \tag{C.13}$$

Note that $\mathrm{GAP}(\psi) \geq 0$ based on the definition, and for all $\psi \subseteq \Phi$. Moreover, according to assumption there exist $\psi^* \subset \Phi$, such that $\mathrm{GAP}(\psi^*) > 0$. Let us define $\Gamma^*$ as

$$\Gamma^* \triangleq \sup_{\lambda \in \mathbb{R} \cup \{\pm\infty\}} \sup_{\psi \subseteq \Phi \mid \Lambda(\psi) \leq \lambda} \Gamma(\psi; \lambda). \tag{C.14}$$

It is easy to see that $\Gamma^* \geq 0$, since $\psi = \Phi - \phi^*$ and $\lambda \geq \Lambda(\Phi - \phi^*)$ lead to $\Gamma(\psi, \lambda) = 0$. The rest of the proof can be divided into two separate parts, based on the assumptions on the value of $\Gamma^*$ w.r.t. function set $\Phi$, and probability distribution $P_0$. First, assume $\Gamma^* > 0$. Then, it can be easily checked that there exists $\zeta > 0$, $\lambda \in \mathbb{R}$ and $\psi \subset \Phi$ such that for any $\eta \in [0, 1]$:

$$\mathrm{GAP}(\psi) + (1 - \eta)\Gamma(\psi; \lambda) - \zeta \geq 0. \tag{C.15}$$

In the second regime, we assume $\Gamma^* = 0$. This very special case indicates a highly incompatible pair of hypothesis set $\Phi$ and distribution $P_0$. In simple words, it means there are functions such as $\phi_{\mathrm{inc}} \in \Phi$, so $\phi_{\mathrm{inc}}$ is highly correlated with label-conditional distribution $P_{0|\boldsymbol{x}}$, in an expected sense. Therefore, it produces large expected loss values, while it can easily fool the learner during the pseudo-labeling procedure (for example, by assigning very small loss values for some irrelevant labels). In this case, $\lambda = +\infty$ (which means $\psi = \Phi$) gives us the desired result and completes the proof. $\qquad\square$

Based on Definition C.4 and previous discussions, the following theorem bounds the true expected adversarial risk, i.e. $\mathbb{E}_{P_0}\{\phi_\gamma(\boldsymbol{Z}; \theta)\}$ based on the expected value of the proposed risk $\mathbb{E}_{P_0}\left\{\hat{R}_{\mathrm{SSAR}}(\theta; \boldsymbol{D})\right\}$, for all $\theta$ that happen to be in a neighborhood of its minimizer.

**Theorem C.1** (Statistical Consistency). *Assume the function set* $\Phi \triangleq \{\phi_\gamma(\cdot; \theta) \mid \theta \in \Theta\}$ *of adversarial loss functions* $\phi_\gamma : \mathcal{Z} \times \Theta \to \mathbb{R}$ *defined in* (4), *for a feature-label space* $\mathcal{Z} \triangleq \mathcal{X} \times \mathcal{Y}$, *a parameter space* $\Theta$ *and dual parameter* $\gamma \geq 0$. *Let* $P_0 \in M(\mathcal{X} \times \mathcal{Y})$ *to be any distribution. Also, assume* $\theta^*$ *to be the minimizer of the actual adversarial loss, i.e.* $\theta^* = \operatorname{argmin}_{\theta \in \Theta} \mathbb{E}_{P_0}\{\phi_\gamma(\boldsymbol{Z}; \theta)\}$. *Let* $\eta \in [0, 1]$ *to denote a supervision ratio, and assume* $\zeta \geq 0$ *and* $\lambda \in \mathbb{R} \cup \{\pm\infty\}$ *such that the following condition holds:*

$$\eta \geq \mathrm{MSR}_{(\Phi, P_0)}(\lambda, \zeta). \tag{C.16}$$

*Consider a partially labeled dataset* $\boldsymbol{D} \triangleq \{(\boldsymbol{X}_i, y_i)\}_{i=1}^n$ *consisting of* $n$ *i.i.d. samples drawn from* $P_0$, *where labels can be observed with probability of* $\eta$, *independently from each other. Then, there*

*exists a neighborhood $\Theta_{\mathrm{local}}$, such that $\theta^* \in \Theta_{\mathrm{local}} \subseteq \Theta$ and all the following relations hold:*

$$\operatorname*{argmin}_{\theta \in \Theta} \mathbb{E}_{P_0} \left\{ \hat{R}_{\mathrm{SSAR}} \left( \theta; \boldsymbol{D} \right) \right\} \in \Theta_{\mathrm{local}},$$

$$\mathbb{E}_{P_0} \left\{ \hat{R}_{\mathrm{SSAR}} \left( \theta; \boldsymbol{D} \right) - \hat{R}_{\mathrm{SSAR}} \left( \theta^*; \boldsymbol{D} \right) \right\} \geq \zeta, \qquad \forall \theta \notin \Theta_{\mathrm{local}},$$

$$and \quad \mathbb{E}_{P_0} \left\{ \phi_\gamma \left( \boldsymbol{Z}; \theta \right) \right\} + \gamma\epsilon \leq \mathbb{E}_{P_0} \left\{ \hat{R}_{\mathrm{SSAR}} \left( \theta; \boldsymbol{D} \right) \right\}, \qquad \forall \theta \in \Theta_{\mathrm{local}}, \tag{C.17}$$

*where the term $\gamma\epsilon$ appears due to the definition of $\hat{R}_{\mathrm{SSAR}}$ in Theorem 1.*

*Proof.* Based on the proof of Lemma C.2 and definition of $\mathrm{MSR}_{(\Phi, P_0)}$, it can be easily checked that the condition $\eta \geq \mathrm{MSR}_{(\Phi, P_0)} \left( \lambda, \zeta \right)$ implies that:

$$\exists \psi \subseteq \Phi \left| \begin{array}{l} \dfrac{\mathrm{GAP} \left( \psi \right) - \zeta}{1 - \eta} + \Gamma \left( \psi; \lambda \right) \geq 0 \quad \text{and} \quad \rho_\lambda \left( \phi \right) \geq 0, \ \forall \phi \in \psi. \end{array} \right. \tag{C.18}$$

Let $\phi^* \triangleq \phi_\gamma \left( \cdot; \theta^* \right)$. According to the definition of GAP and $\Gamma$ in Definition C.3, the first condition in the above results in the following chain of relations:

$$\begin{aligned}
\zeta &\leq \min_{\phi \in \Phi - \psi} \mathbb{E}_{P_0} \left\{ \phi - \phi^* \right\} + \left( 1 - \eta \right) \min_{\phi \in \Phi - \psi} \mathbb{E}_{P_{0\boldsymbol{x}}} \left\{ \rho_\lambda \left( \phi \right) - \rho_\lambda \left( \phi^* \right) \right\} \\
&\leq \min_{\phi \in \Phi - \psi} \left\{ \mathbb{E}_{P_0} \left\{ \phi \right\} + \left( 1 - \eta \right) \rho_\lambda \left( \phi \right) \right\} - \left\{ \mathbb{E}_{P_0} \left\{ \phi^* \right\} - \left( 1 - \eta \right) \rho_\lambda \left( \phi^* \right) \right\} \\
&= \min_{\phi \in \Phi - \psi} \mathbb{E}_{P_0} \left\{ \eta\phi + \left( 1 - \eta \right) \operatorname*{softmin}_{y \in \mathcal{Y}}^{(\lambda)} \left\{ \phi_{\boldsymbol{X}} \right\} \right\} - \mathbb{E}_{P_0} \left\{ \eta\phi^* + \left( 1 - \eta \right) \operatorname*{softmin}_{y \in \mathcal{Y}}^{(\lambda)} \left\{ \phi^*_{\boldsymbol{X}} \right\} \right\} \\
&= \min_{\theta \in \Theta - \Theta_{\mathrm{local}}} \mathbb{E}_{P_0} \left\{ \hat{R}_{\mathrm{SSAR}} \left( \theta; \boldsymbol{D} \right) \right\} - \mathbb{E}_{P_0} \left\{ \hat{R}_{\mathrm{SSAR}} \left( \theta^*; \boldsymbol{D} \right) \right\}, \tag{C.19}
\end{aligned}$$

where $\Theta_{\mathrm{local}}$ denotes the subset of parameter space $\Theta$ that corresponds to function subset $\psi$. This proves the first two arguments of the Theorem. Note that the first argument can be directly deduced from the second one, and we have only written it separately for the sake of emphasis and clarity. The third argument can also be directly deduced from the fact that $\Lambda \left( \psi \right) \leq \lambda$. Note that based on Definition C.3 and for all $\phi \in \psi$ (or equivalently $\theta \in \Theta_{\mathrm{local}}$), we have $\rho_{\Lambda(\psi)} \left( \phi \right) \geq 0$. Therefore:

$$\begin{aligned}
\left( 1 - \eta \right) \rho_{\Lambda(\psi)} \left( \phi \right) &= \left( 1 - \eta \right) \mathbb{E}_{P_{0\boldsymbol{x}}} \left\{ \operatorname*{softmin}_{y \in \mathcal{Y}}^{(\Lambda(\psi))} \left\{ \phi_{\boldsymbol{X}} \right\} - \mathbb{E}_{P_{0|\boldsymbol{x}}} \left\{ \phi_{\boldsymbol{X}} \right\} \right\} \\
&= \left( 1 - \eta \right) \mathbb{E}_{P_{0\boldsymbol{x}}} \left\{ \operatorname*{softmin}_{y \in \mathcal{Y}}^{(\Lambda(\psi))} \left\{ \phi_{\boldsymbol{X}} \right\} \right\} - \left( 1 - \eta \right) \mathbb{E}_{P_0} \left\{ \phi \right\} \\
&= \eta \mathbb{E}_{P_0} \left\{ \phi \right\} + \left( 1 - \eta \right) \mathbb{E}_{P_{0\boldsymbol{x}}} \left\{ \operatorname*{softmin}_{y \in \mathcal{Y}}^{(\Lambda(\psi))} \left\{ \phi_{\boldsymbol{X}} \right\} \right\} - \mathbb{E}_{P_0} \left\{ \phi \right\} \\
&= \mathbb{E}_{P_0} \left\{ \hat{R}_{\mathrm{SSAR}} \left( \theta; \boldsymbol{D} \right) \right\} - \mathbb{E}_{P_0} \left\{ \phi_\gamma \left( \boldsymbol{Z}; \theta \right) \right\} - \gamma\epsilon \geq 0. \tag{C.20}
\end{aligned}$$

Taking into account the fact that $\operatorname*{softmin}_{y \in \mathcal{Y}}^{(\lambda)} \left( \cdot \right)$ is an increasing function w.r.t. $\lambda$ leads to the third argument, and thus completes the proof. $\square$

Theorem C.1 provides a mathematical foundation for establishing a general learning-theoretic bound on the generalization aspect of self-learning paradigm, that can be applied to our distributionally robust setting as well. Intuitively, it states that for good choices of the pair $(\eta, \lambda)$, one can guarantee the following two outcomes:

First, the minimizer of the expected proposed loss happens to be in a neighborhood of the true minimizer, i.e. $\operatorname{argmin}_{\theta \in \Theta} \sup_{P \in \mathcal{B}_\epsilon(P_0)} \mathbb{E}_P \left\{ \ell \left( \boldsymbol{Z}; \theta \right) \right\}$. Also, a positive margin $\zeta > 0$ can be considered which puts a gap between the minimum value of the proposed expected loss and those that fall outside of this neighborhood. This margin will be extremely helpful when we are dealing with empirical risks instead of the statistical ones (see the proof of Theorem 3).

Second, all over the above-mentioned neighborhood, $R_{\mathrm{SSAR}}$ provides an upper-bound on the true expected loss. In this regard, as long as a minimum level of pessimism is considered with respect to the compatibility of the hypothesis set and distribution duo, i.e. $\lambda \geq \Lambda(\psi)$, it can be guaranteed that the self-learning module does not overfit and assigns meaningful labels to the unlabeled data.

From a more practical perspective, the mathematical formulation of MSR function in Definition C.4 may seem too implicit to be applicable in real-world problems. To show the usefulness of this measure, Lemma C.3 analytically computes $\mathrm{MSR}_{(\Phi, P_0)}$ for any pair $(\Phi, P_0)$ that satisfies a strong *cluster assumption*. In particular, we show that by using Definition C.4 followed by some simple algebra, one can reattain a previously established generalization bound for the case of cluster assumption.

**Lemma C.3.** *Assume $\Phi \subseteq \mathbb{R}^{\mathcal{X}}$ and data distribution $P_0 \in M(\mathcal{X} \times \mathcal{Y})$ that satisfies a strong cluster assumption. Therefore, $P_0$ is a mixture of two distributions with non-overlapping supports over $\mathcal{X}$, where mixture components only correspond to $y = -1$ and $y = +1$, respectively. Let $\Phi$ be associated to a family of binary classifiers, where for each $\phi \in \Phi$ we have $\phi(X, y) = \infty \cdot \phi_{acc}(X, y) + \phi_{mar}(X)$. In this regard, $\phi_{acc} \in \{0, 1\}$ checks if the label $y$ matches with $X$ w.r.t. $\phi$, and $\phi_{mar}(X) \in \mathbb{R}$ penalizes the margin of $X$, i.e. distance of $X$ from the classifier's boundary. Then, for a sufficiently small $\zeta > 0$, we have $\mathrm{MSR}_{(\Phi, P_0)}(\lambda, \zeta) = 0$ for any $\lambda \in \mathbb{R} \cup \pm\infty$.*

*Proof.* Let $\psi \subseteq \Phi$ be a subset of classifiers that classify all the data samples correctly, i.e.

$$\forall \phi \in \psi \ \Rightarrow \ \mathbb{E}_{P_0}\{\phi_{acc}\} = 0.$$

However, classifiers in $\psi$ may have different expected margins. Also, assume the optimal classifier or equivalently the minimizer of empirical risk minimization, denoted by $\phi^*$, is also inside $\psi$. Then, some simple calculations reveal that for every $\phi \in \psi$ and any $\lambda$ we have $\rho_\lambda(\phi) = 0$ which means $\Lambda(\psi) = -\infty$. Also, we have $\Gamma(\psi; \lambda) \geq 0$, again for any $\lambda$, while $\mathrm{GAP}(\psi)$ is strictly positive for any non-trivial choice of $\Phi$. The latter is due to the fact that $\phi^* \in \psi$.

As long as $\Phi$ is assumed to be a learnable family of binary classifiers with a bounded VC-dimension, we have $\mathcal{R}_{n,(\epsilon,\eta)}^{(\mathrm{SSM})}(\Phi) = O(n^{-1/2})$ due to Lemma E.3. Recalling the generalization bound of Theorem 3, this alternatively means that we can have $\zeta = O(n^{-1/2})$. Then, for a sufficiently large $n$, $\mathrm{MSR}_{\Phi, P_0}(\lambda, O(n^{-1/2}))$ becomes zero for any $\lambda \in \mathbb{R} \cup \pm\infty$. This result is in full agreement with the previous bounds that are specifically derived for generic learnability of statistical models when the strong (non-overlapping) form of cluster assumption holds. Note that for absolute learnability, at least one data point with a label is needed to decide which cluster is which. $\qquad \square$

This result indicates that for a fairly large $n$, the generalization bound of Theorem 3 holds for any supervision ratio, as long as there exists only one labeled sample. As it is evident from the proof of Lemma C.3, this generalization bound has been achieved with far less effort compared to the previous studies on this particular problem. This also suggests that many existing theoretical frameworks in semi-supervised learning can be potentially considered as special cases of the proposed setting.

# D  Auxiliary Theorems and Proofs

*Proof of Theorem 1.* The proof proceeds by the substitution of original proposed semi-supervised problem in (2) by its dual form. This way, we can take advantage of the good mathematical properties that this dual form can provide, specially w.r.t. maximization over $P \in \mathcal{B}_\epsilon(S)$. The following lemma (see Theorem 1 and Remark 1 of [8]), formulates the dual form:

**Lemma D.1** (Lagrangian Relaxation and Duality)**.** *Assume $\mathcal{Z}$ to be a sample space and let $\Theta$ to denote the space of parameters. Let loss function $\ell : \mathcal{Z} \times \Theta \to \mathbb{R}_{\geq 0}$ and function $c : \mathcal{Z} \times \mathcal{Z} \to \mathbb{R}_{\geq 0}$ to be continuous, and further assume $c$ is lower semi-continuous and $c(z, z) = 0$, $\forall z \in \mathcal{Z}$. Then, for any $\epsilon \geq 0$ and any distribution $Q \in M(\mathcal{Z})$, the following equality holds for all $\theta \in \Theta$:*

$$\sup_{P \in \mathcal{B}_\epsilon(Q)} \mathbb{E}_P\{\ell(Z; \theta)\} = \inf_{\gamma \geq 0}\left\{\gamma\epsilon + \mathbb{E}_Q\left\{\sup_{z' \in \mathcal{Z}} \ell(z'; \theta) - \gamma c(z', Z)\right\}\right\}. \tag{D.1}$$

Proof is explained in details in the original reference. Based on the duality equation in Lemma D.1, the following chain of relations hold:

$$\inf_{S\in\hat{\mathcal{P}}(\boldsymbol{D})}\left(\sup_{P\in\mathcal{B}_\epsilon(S)}\mathbb{E}_P\left\{\ell\left(\boldsymbol{X},y;\theta\right)\right\}+\frac{1}{\lambda}\left(\frac{n_{\mathrm{ul}}}{n}\right)\hat{\mathbb{E}}_{\boldsymbol{D}_{\mathrm{ul}}}\left\{\mathbb{H}\left(S_{|\boldsymbol{X}}\right)\right\}\right) \tag{D.2}$$

$$=\inf_{S\in\hat{\mathcal{P}}(\boldsymbol{D})}\left[\inf_{\gamma\geq0}\left(\gamma\epsilon+\mathbb{E}_S\left\{\sup_{\boldsymbol{z}'\in\mathcal{Z}}\ell\left(\boldsymbol{z}';\theta\right)-\gamma c\left(\boldsymbol{z}',\boldsymbol{Z}\right)\right\}\right)+\frac{1}{\lambda}\left(\frac{n_{\mathrm{ul}}}{n}\right)\hat{\mathbb{E}}_{\boldsymbol{D}_{\mathrm{ul}}}\left\{\mathbb{H}\left(S_{|\boldsymbol{X}}\right)\right\}\right]$$

$$=\inf_{\gamma\geq0}\left[\gamma\epsilon+\inf_{S\in\hat{\mathcal{P}}(\boldsymbol{D})}\left(\mathbb{E}_S\left\{\sup_{\boldsymbol{z}'\in\mathcal{Z}}\ell\left(\boldsymbol{z}';\theta\right)-\gamma c\left(\boldsymbol{z}',\left(\boldsymbol{X},y\right)\right)\right\}+\frac{1}{\lambda}\left(\frac{n_{\mathrm{ul}}}{n}\right)\hat{\mathbb{E}}_{\boldsymbol{D}_{\mathrm{ul}}}\left\{\mathbb{H}\left(S_{|\boldsymbol{X}}\right)\right\}\right)\right]$$

$$=\inf_{\gamma\geq0}\left[\gamma\epsilon+\left(\frac{n_{\mathrm{l}}}{n}\right)\frac{1}{n_{\mathrm{l}}}\sum_{i\in\mathcal{I}_{\mathrm{l}}}\left(\sup_{\boldsymbol{z}'\in\mathcal{Z}}\ell\left(\boldsymbol{z}';\theta\right)-\gamma c\left(\boldsymbol{z}',\left(\boldsymbol{X}_i,y_i\right)\right)\right)\right.$$

$$\left.+\left(\frac{n_{\mathrm{ul}}}{n}\right)\frac{1}{n_{\mathrm{ul}}}\sum_{i\in\mathcal{I}_{\mathrm{ul}}}\inf_{\Omega\in M(\mathcal{Y})}\left(\mathbb{E}_\Omega\left\{\sup_{\boldsymbol{z}'\in\mathcal{Z}}\ell\left(\boldsymbol{z}';\theta\right)-\gamma c\left(\boldsymbol{z}',\left(\boldsymbol{X}_i,y\right)\right)\right\}+\frac{1}{\lambda}\mathbb{H}\left(\Omega\right)\right)\right],$$

where the last inequality is a direct result of defining $\hat{\mathcal{P}}\left(\boldsymbol{D}\right)$ in Definition 1. Therefore, each $S\in\hat{\mathcal{P}}\left(\boldsymbol{D}\right)$ can be regarded as a weighted (with weights $n_{\mathrm{l}}/n$ and $n_{\mathrm{ul}}/n$, respectively) mixture of $\mathbb{P}_{\boldsymbol{D}_{\mathrm{l}}}$, i.e. delta-spikes over the labeled samples, and $\mathbb{P}_{\boldsymbol{D}_{\mathrm{ul}}}\tilde{\Omega}$, i.e. the same for unlabeled feature vectors which are multiplied by arbitrary conditional distributions of the form $\Omega\in M^\mathcal{X}\left(\mathcal{Y}\right)$. The two summations above which are over labeled and unlabeled samples, respectively, correspond to this bi-mixture formalism. Thus, the chain of relations in (D.2) can be continued as

$$=\inf_{\gamma\geq0}\left[\gamma\epsilon+\frac{1}{n}\sum_{i\in\mathcal{I}_{\mathrm{l}}}\phi_\gamma\left(\boldsymbol{X}_i,y_i|\theta\right)+\frac{1}{n}\sum_{i\in\mathcal{I}_{\mathrm{ul}}}\left(\inf_{\Omega\in M(\mathcal{Y})}\sum_{y\in\mathcal{Y}}\Omega_y\phi_\gamma\left(\boldsymbol{X}_i,y;\theta\right)+\frac{1}{\lambda}\mathbb{H}\left(\Omega\right)\right)\right]$$

$$=\inf_{\gamma\geq0}\left[\gamma\epsilon+\frac{1}{n}\sum_{i\in\mathcal{I}_{\mathrm{l}}}\phi_\gamma\left(\boldsymbol{X}_i,y_i;\theta\right)+\frac{1}{n}\sum_{i\in\mathcal{I}_{\mathrm{ul}}}\operatorname*{softmin}_{y\in\mathcal{Y}}^{(\lambda)}\left\{\phi_\gamma\left(\boldsymbol{X}_i,y;\theta\right)\right\}\right]+const, \tag{D.3}$$

where $const$ deos not depend on $\gamma$ or $\theta$, and the last equality is due to the following lemma:

**Lemma D.2.** *Assume an arbitrary vector $\boldsymbol{b}\in\mathbb{R}^d$ for $d\in\mathbb{N}$, and also let $\mathcal{F}\triangleq\{1,\ldots,d\}$. Then the following relation holds for all $\lambda\in\mathbb{R}\cup\{\pm\infty\}$:*

$$\operatorname*{softmin}_{i\in\mathcal{F}}^{(\lambda)}\left(b_1,\ldots,b_d\right)=\inf_{\boldsymbol{q}\in M(\mathcal{F})}\boldsymbol{q}^T\boldsymbol{b}+\frac{1}{\lambda}\mathbb{H}\left(\boldsymbol{q}\right)-\frac{1}{\lambda}\log d, \tag{D.4}$$

*where $\mathbb{H}\left(\cdot\right)$ denotes the Shannon entropy of distribution $\boldsymbol{q}$ over $\mathcal{F}$.*

*Proof.* The main idea is to replace the term $\boldsymbol{q}^T\boldsymbol{b}$ with

$$\boldsymbol{q}^T\boldsymbol{b}=\sum_{i\in\mathcal{F}}q_ib_i=\frac{1}{\lambda}\sum_{i\in\mathcal{F}}q_i\log e^{\lambda b_i}. \tag{D.5}$$

Also, note that $\frac{1}{\lambda}\mathbb{H}\left(\boldsymbol{q}\right)-\frac{1}{\lambda}\log d=\frac{-1}{\lambda}\mathcal{D}_{\mathrm{KL}}\left(\boldsymbol{q}\|\mathcal{U}\right)$, where $\mathcal{D}_{\mathrm{KL}}$ is the Kullback–Leibler divergence between two probability measures and $\mathcal{U}\in M\left(\mathcal{F}\right)$ denotes the uniform measure on $\mathcal{F}$. As a result, the overall objective function can be rewritten as

$$\boldsymbol{q}^T\boldsymbol{b}-\frac{1}{\lambda}\mathcal{D}_{\mathrm{KL}}\left(\boldsymbol{q}\|\mathcal{U}\right)$$

$$=-\frac{1}{\lambda}\mathcal{D}_{\mathrm{KL}}\left(\boldsymbol{q}\|\mathcal{U}\right)+\frac{1}{\lambda}\sum_{i\in\mathcal{F}}q_i\log e^{\lambda b_i}$$

$$=-\frac{1}{\lambda}\sum_{i\in\mathcal{F}}q_i\log\left(dq_i\right)+\frac{1}{\lambda}\sum_{i\in\mathcal{F}}q_i\log e^{\lambda b_i}$$

$$=-\frac{1}{\lambda}\sum_{i\in\mathcal{F}}q_i\log\left(\frac{q_i}{\frac{1}{d}e^{\lambda b_i}}\right)=-\frac{1}{\lambda}\sum_{i\in\mathcal{F}}q_i\log\left(\frac{q_i}{\frac{\alpha}{d}e^{\lambda b_i}}\right)-\frac{1}{\lambda}\log\alpha,$$

for all $\alpha > 0$. Then, it can be readily verified that by setting $\alpha^{-1} \triangleq \frac{1}{d} \sum_{i \in \mathcal{F}} e^{\lambda b_i}$, the optimization problem in lemma becomes

$$\inf_{\boldsymbol{q} \in M(\mathcal{F})} -\frac{1}{\lambda} \mathcal{D}_{\mathrm{KL}} \left( q_i \middle\| \frac{\alpha}{d} e^{\lambda b_i} \right) + \frac{1}{\lambda} \log \left( \frac{1}{d} \sum_{i \in \mathcal{F}} e^{\lambda b_i} \right), \tag{D.6}$$

whose solution always happens to be $q_i^* = \frac{\alpha}{d} e^{-b_i/\lambda}$, regardless of the sign of $\lambda$. Therefore, the solution of the primary optimization problem in lemma would be

$$\frac{1}{\lambda} \log \left( \frac{1}{d} \sum_{i \in \mathcal{F}} e^{\lambda b_i} \right) = \operatorname*{softmin}_{i \in \mathcal{F}}^{(\lambda)} (b_1, \ldots, b_d), \tag{D.7}$$

which completes the proof. $\qquad \square$

According to the duality relation between $\gamma$ and $\epsilon$, the minimization over $\gamma$ is not necessary in almost all practical situations, where the same methodologies for evaluating a *practically good* value for $\epsilon$, such as cross-validation, can be used for $\gamma$ as well. $\qquad \square$

*Proof of Theorem 2.* The proof is based on a number of techniques used in [9], and can be considered as a generalization of Theorem 2 of [1] for the semi-supervised settings. Similarly, let us define the following set of Lipschitz constants, based on the smoothness constraints assumed in Theorem 2:

$$\|\nabla_\theta \ell(\boldsymbol{z}; \theta) - \nabla_\theta \ell(\boldsymbol{z}; \theta')\|_* \le L_{\theta\theta} \|\theta - \theta'\|, \qquad \|\nabla_\theta \ell(\boldsymbol{z}; \theta) - \nabla_\theta \ell(\boldsymbol{z}'; \theta)\|_* \le L_{\theta\boldsymbol{z}} \|\boldsymbol{z} - \boldsymbol{z}'\|,$$

$$\|\nabla_{\boldsymbol{z}} \ell(\boldsymbol{z}; \theta) - \nabla_{\boldsymbol{z}} \ell(\boldsymbol{z}; \theta')\|_* \le L_{\boldsymbol{z}\theta} \|\theta - \theta'\|, \qquad \|\nabla_{\boldsymbol{z}} \ell(\boldsymbol{z}; \theta) - \nabla_{\boldsymbol{z}} \ell(\boldsymbol{z}'; \theta)\|_* \le L_{\boldsymbol{z}\boldsymbol{z}} \|\boldsymbol{z} - \boldsymbol{z}'\|,$$

where $\{L_{\theta\theta}, L_{\theta\boldsymbol{z}}, L_{\boldsymbol{z}\theta}, L_{\boldsymbol{z}\boldsymbol{z}}\}$ are a set of Lipschitz constants, $\|\cdot\|$ can be any valid norm (generally different norms should be used for $\mathcal{Z}$ and $\Theta$) and $\|\cdot\|_*$ denotes the corresponding dual norm(s). Also, the inequalities should hold for all $\boldsymbol{z}, \boldsymbol{z}' \in \mathcal{Z}$ and all $\theta, \theta' \in \Theta$.

In our case, i.e. a semi-supervised setting, one also needs to show that $\nabla_\theta \operatorname*{softmin}_{y \in \mathcal{Y}}^{(\lambda)} \{\phi_\gamma(\boldsymbol{z}; \cdot)\}$ is Lipschitz with respect to $\theta$, for all $\boldsymbol{z} \in \mathcal{Z}$. Before that, Lemma D.3 shows that under the above-mentioned constraints on the Lipschitz-ness of gradients of $\ell$, $\phi_\gamma(\boldsymbol{z}; \theta)$ also has Lipschitz gradients.

**Lemma D.3.** *Assume $\ell : \mathcal{Z} \times \Theta \to \mathbb{R}_{\ge 0}$ is smooth and universally differentiable w.r.t. its input arguments. Also assume $\ell$ has Lipschitz gradients with constants $\{L_{\theta\theta}, L_{\theta\boldsymbol{z}}, L_{\boldsymbol{z}\theta}, L_{\boldsymbol{z}\boldsymbol{z}}\}$, for any fixed norm $\|\cdot\|$. Also, assume a transportation cost $c$, which has the properties of Lemma E.1. Then, the following Lipschit-ness property holds for gradients of $\phi_\gamma(\boldsymbol{z}; \theta) = \sup_{\boldsymbol{z}' \in \mathcal{Z}} \ell(\boldsymbol{z}'; \theta) - \gamma c(\boldsymbol{z}', \boldsymbol{z})$:*

$$\|\nabla_\theta \phi_\gamma(\boldsymbol{z}; \theta) - \nabla_\theta \phi_\gamma(\boldsymbol{z}; \theta')\|_* \le \left( L_{\theta\theta} + \frac{L_{\boldsymbol{z}\theta} L_{\theta\boldsymbol{z}}}{\gamma - L_{\boldsymbol{z}\boldsymbol{z}}} \right) \|\theta - \theta'\|, \quad \forall \theta, \theta' \in \Theta, \tag{D.8}$$

*for all $\gamma > L_{\boldsymbol{z}\boldsymbol{z}}$.*

For proof of Lemma D.3, see Lemma 1 of [1]. Based on this result, the following lemma provides Lipschitz constants for the softmin operator over a finite number of $\phi_\gamma(\cdot; \cdot)$ functions, for any $\lambda \in \mathbb{R}$.

**Lemma D.4.** *For a feature-label space $\mathcal{Z} = \mathcal{X} \times \mathcal{Y}$, assume loss function $\ell : \mathcal{Z} \times \Theta \to \mathbb{R}_{\ge 0}$, transportation cost $c$ and the resulting adversarial loss $\phi_\gamma(\cdot; \cdot) : \mathcal{Z} \times \Theta \to \mathbb{R}$ with $\gamma > L_{\boldsymbol{z}\boldsymbol{z}}$, such that all satisfy the constraints of Lemma D.3. Also, assume there exists $\sigma \ge 0$ such that $\|\nabla_\theta \ell(\boldsymbol{z}; \theta)\| \le \sigma$ for all $\theta \in \Theta$. Then, for all $\lambda \in \mathbb{R}$, the following Lipschitz-ness property holds:*

$$\left\| \nabla_\theta \operatorname*{softmin}_{y \in \mathcal{Y}}^{(\lambda)} \{\phi_\gamma(\boldsymbol{Z}; \theta)\} - \nabla_\theta \operatorname*{softmin}_{y \in \mathcal{Y}}^{(\lambda)} \{\phi_\gamma(\boldsymbol{Z}; \theta')\} \right\|_* \le \left( L_{\theta\theta} + \frac{L_{\boldsymbol{z}\theta} L_{\theta\boldsymbol{z}}}{\gamma - L_{\boldsymbol{z}\boldsymbol{z}}} + 2\sigma^2 |\lambda| |\mathcal{Y}| \right) \|\theta - \theta'\|, \tag{D.9}$$

*for all $\boldsymbol{Z} \in \mathcal{Z}$ and $\theta, \theta' \in \Theta$.*

In order to avoid discontinuity in the proof, the proof of Lemma D.4 is presented in Appendix E instead of here. Also, let $B \triangleq \frac{1}{2} \left( L_{\theta\theta} + \frac{L_{\boldsymbol{z}\theta} L_{\theta\boldsymbol{z}}}{\gamma - L_{\boldsymbol{z}\boldsymbol{z}}} \right)$, where $B$ represents one of the constants mentioned in Theorem 2.

The last lemma which is needed to finalize the proof of Theorem 2 aims to bound the maximum discrepancy that one might observe, given that the inner maximization in (E.2) (corresponds to line 6 of Algorithm 1) is solved up to an approximation error of $\delta > 0$.

**Lemma D.5.** *Assume $\hat{z}^* \in \mathcal{Z}$ to be a $\delta$-approximate maximizer of* (E.2) *for the input $z_0 \in \mathcal{Z}$, loss function $\ell$, and transportation cost $c$. Let the consequent adversarial loss function $\phi_\gamma$ to satisfy all the constraints mentioned in Lemma D.3 in addition to $\gamma > L_{zz}$. Then, the following upper-bound holds for all $z_0 \in \mathcal{Z}$:*

$$\left\| \nabla_\theta \phi_\gamma \left( z_0; \theta \right) - \nabla_\theta \ell \left( \hat{z}^*; \theta \right) \right\|_*^2 \leq \frac{L_{z\theta} L_{\theta z}}{\gamma - L_{zz}} \delta. \tag{D.10}$$

Proof of Lemma D.5 is given in Appendix E. Also, Let $C \triangleq \frac{L_{z\theta} L_{\theta z}}{\gamma - L_{zz}}$, recalling $C$ as another constant mentioned in Theorem 2.

Algorithm 1 for a mini-batch size of $k = 1$ picks one data-point randomly from $D$ at each iteration. Also, data points at $D$ are assumed to be drawn independently from an unknown but fixed distribution $P_0$. Therefore, one can consider a two-step data generation model in order to analyze the semi-supervised stochastic gradient descent as follows:

- $\mathcal{O}$ (*Observation step*): Draw a bi-categorical random variable (denoted as observation variable) $h \in \mathcal{H} \triangleq \{\mathrm{l}, \mathrm{ul}\}$, with probabilities $n_\mathrm{l}/n$ and $n_\mathrm{ul}/n$ for labeled and unlabeled categories, respectively.

- $\mathcal{G}$ (*Generation step*): Conditioned on $h$, draw a sample from $P_0$ if $h = \mathrm{l}$, and from $P_{0,X}$ if $h = \mathrm{ul}$.

Consider a coupled first-order *Markov stochastic process* defined as $(h_0, \theta_0), \ldots, (h_T, \theta_T)$, where $h_i$s denote the observation variables and $\theta_i$s are the consequent outputs of Algorithm 1 after $T$ iterations. Here, $\theta_0$ can have any initial distribution over $\Theta$. Using the techniques reviewed in [9] (also similar to Theorem 2 of [1]), the following result holds for for $1 < t \leq T$:

$$\mathbb{E}_\mathcal{G} \left\{ \hat{R}_{\mathrm{SSAR}} \left( \theta_{t+1}; D \right) - \hat{R}_{\mathrm{SSAR}} \left( \theta_t; D \right) \big| \theta_t, h_t \right\} \leq - \alpha \left( \frac{1}{2} - \alpha L_{h_t} \right) \left\| \nabla_\theta \hat{R}_{\mathrm{SSAR}} \left( \theta^t | D \right) \right\|_2^2$$
$$+ \frac{1}{2} \left( \alpha + 5\alpha^2 L_{h_t} \right) C\delta + \frac{1}{2} \alpha^2 \sigma^2 L_{h_t}, \tag{D.11}$$

where $\mathbb{E}_\mathcal{G}$ refers to expectation w.r.t. the randomness of dataset $D$, and given that the information about each sample is labeled or not is known. Also, $L_h \in \mathbb{R}_{\geq 0}^\mathcal{H}$ denotes the Lipschitz constants for the gradients (w.r.t. $\theta \in \Theta$) of the loss summands in (3). Based on Lemma D.4, we have

$$L_h \leq \begin{cases} 2 \left( B + \sigma^2 |\lambda| |\mathcal{Y}| \right) & h = \mathrm{ul} \\ 2B & h = \mathrm{l} \end{cases} . \tag{D.12}$$

Now, it should be noted that $\mathbb{E}_{\mathrm{total}} \{\cdot\} = \mathbb{E}_\mathcal{O} \{\mathbb{E}_\mathcal{G} \{\cdot | h \in \mathcal{H}\}\}$, where $\mathbb{E}_{\mathrm{total}}$ denotes the total expectation which is w.r.t. the dataset $D$ whose samples are drawn i.i.d. from $P_0$ and also the randomness of SGD used in Algorithm 1. Also, due to the independence assumption on observing each label with probability $\eta$, we have

$$\mathbb{E}_\mathcal{O} \{L_{h_t}\} = 2 \left( B + \bar{\eta} \sigma^2 |\lambda| |\mathcal{Y}| \right) , \forall t. \tag{D.13}$$

Combining the above arguments with (D.11) directly leads us to the claims in Theorem 2 and completes the proof. $\qquad\square$

**Theorem D.1** (Convergence of hard decisions, $\lambda = \pm\infty$). *Consider the setting described in Theorem 2, where $\ell$ is twice differentiable w.r.t. $\theta$ all over $\mathcal{Z} \times \Theta$. Assume one sets $\lambda = +\infty$ or $\lambda = -\infty$. Also, assume step-size $\alpha$ and approximation interval $\delta$ in Algorithm 1 can change during the iterations. Then, there exist a sequence of step-sizes $\alpha_1, \alpha_2, \ldots$ and a sequence of approximation intervals $\delta_1, \delta_2, \ldots$ for which Algorithm 1 converges to a local minimizer of $\hat{R}_{\mathrm{SSAR}} (\theta; D)$, as $T \to \infty$ where $T$ is the number of iterations.*

*Proof.* Problem setting for $\lambda = +\infty$ results into a minimax problem, i.e. minimizing over $\theta \in \Theta$ while maximizing over $y_i \in \mathcal{Y}$, $i \in \mathcal{I}_{\mathrm{ul}}$ for any given $\theta$. Thus, the solution is a local saddle point in $\Theta \times \mathcal{Y}^{|\mathcal{I}_{\mathrm{ul}}|}$. Convergence of combinatoric optimization schemes for such problems are already established (see [7] and [10]), and we avoid to repeat them here.

For the case of $\lambda = -\infty$, we show that by choosing sufficiently small values for $\alpha_i$ and $\delta_i$ for $i = 1, 2, \ldots$, the objective of the optimization always decreases, and thus convergence to a stable point is guaranteed. First, let us define

$$y_i^* (\theta) \triangleq \underset{y \in \mathcal{Y}}{\arg\min} \, \phi_\gamma \left( \boldsymbol{X}_i, y; \theta \right), \tag{D.14}$$

for $i \in \mathcal{I}_{\text{ul}}$. Whenever there are more than one minimizers, one of them is chosen at random. Assume iteration steps $t_s$ and $t_f$ (with $t_s \leq t_f$), such that $y_i^* (\theta_t)$ for $t_s \leq t \leq t_f$ does not change for any $i \in \mathcal{I}_{\text{ul}}$. Then, Algorithm 1 for this period acts exactly like a fully-supervised Stochastic Gradient Descent method on the dataset $\{(\boldsymbol{X}_i, y_i), \, i \in \mathcal{I}_l\} \cup \{(\boldsymbol{X}_i, y_i^* (\theta_t)), \, i \in \mathcal{I}_{\text{ul}}\}$. Consider the set of Lipschitz constants from Theorem 2 (refer to its proof in Appendix D), i.e. $\{L_{\theta\theta}, L_{\theta\boldsymbol{z}}, L_{\boldsymbol{z}\theta}, L_{\boldsymbol{z}\boldsymbol{z}}\}$. Let

$$\delta_t \leq \frac{\gamma - L_{\boldsymbol{z}\boldsymbol{z}}}{2n L_{\theta\boldsymbol{z}} L_{\boldsymbol{z}\theta}} \min_{i=1,2,\ldots,n} \|\nabla_\theta \phi_\gamma \left( \boldsymbol{Z}_i; \theta_{t-1} \right)\|_2, \tag{D.15}$$

where $\boldsymbol{Z}_i = (\boldsymbol{X}_i, y_i), \, i \in \mathcal{I}_l$ and $\boldsymbol{Z}_i = (\boldsymbol{X}_i, y_i^* (\theta_t)), \, i \in \mathcal{I}_{\text{ul}}$. Also assume

$$\alpha_t \leq \min_{i=1,2,\ldots,n} \, \inf_{\theta \in \Theta} \, \frac{4}{9} \left| \lambda_{\max}^{-1} \left\{ \nabla_{\theta\theta}^2 \phi_\gamma \left( \boldsymbol{Z}_i; \theta \right) \right\} \right|, \tag{D.16}$$

where $\nabla_{\theta\theta}^2$ indicates the Hessian matrix operator, and $\lambda_{\max} \{\cdot\}$ extracts the maximum eigenvalue. Then, it can be easily checked that $\phi_\gamma \left( \boldsymbol{Z}_i; \theta_t \right) \leq \phi_\gamma \left( \boldsymbol{Z}_i; \theta_{t-1} \right)$ for all $i = 1, 2, \ldots, n$. This result is due to the fact that for any twice differentiable function $f : \mathbb{R}^d \to \mathbb{R}$, with $\boldsymbol{x}, \boldsymbol{v} \in \mathbb{R}^d$ and $d \in \mathbb{N}$, we have

$$f \left( \boldsymbol{x} + \boldsymbol{v} \right) - f \left( \boldsymbol{x} \right) = \boldsymbol{v}^T \nabla f \left( \boldsymbol{x} \right) + \frac{1}{2} \boldsymbol{v}^T \nabla^2 f \left( \tilde{\boldsymbol{x}} \right) \boldsymbol{v}, \tag{D.17}$$

with $\tilde{\boldsymbol{x}} \in \{\boldsymbol{x} + \mu\boldsymbol{v} | \, 0 \leq \mu \leq 1\}$. Also, based on Lemma D.5 and given the condition on $\delta_t$, we have

$$\Delta \triangleq \frac{\left\| \hat{\partial}_\theta \hat{R}_{\text{SSAR}} \left( \theta_{t-1}; \boldsymbol{D} \right) - \partial_\theta^* \hat{R}_{\text{SSAR}} \left( \theta_{t-1}; \boldsymbol{D} \right) \right\|_2}{\left\| \partial_\theta^* \hat{R}_{\text{SSAR}} \left( \theta_{t-1}; \boldsymbol{D} \right) \right\|_2} \leq \frac{1}{2}, \tag{D.18}$$

where $\hat{\partial}_\theta \hat{R}_{\text{SSAR}} \left( \theta_{t-1}; \boldsymbol{D} \right)$ represents the sub-gradient of $\hat{R}_{\text{SSAR}} \left( \theta_{t-1}; \boldsymbol{D} \right)$ with the inexact solution of (E.2) (a $\delta_t$-approximate solution), while $\partial_\theta^* \hat{R}_{\text{SSAR}} \left( \theta_{t-1}; \boldsymbol{D} \right)$ denotes the exact sub-gradient corresponding to the same data point chosen for iteration $t$. This result holds regardless of the randomness of Algorithm 1 in choosing a sample for computing the sub-gradient. Using (D.17), it can be easily checked that

$$\hat{R}_{\text{SSAR}} \left( \theta_t; \boldsymbol{D} \right) - \hat{R}_{\text{SSAR}} \left( \theta_{t-1}; \boldsymbol{D} \right) \leq$$
$$\left\| \partial_\theta^* \hat{R}_{\text{SSAR}} \left( \theta_{t-1}; \boldsymbol{D} \right) \right\|_2^2 \left( -\alpha_t \left( 1 - \Delta \right) + \frac{\alpha_t^2}{2} \left| \lambda_{\max} \left\{ \nabla_{\theta\theta}^2 \phi_\gamma \left( \boldsymbol{Z}_{\text{chosen}}^{(t)}; \tilde{\theta} \right) \right\} \right| \left( 1 + \Delta \right)^2 \right), \tag{D.19}$$

where $\boldsymbol{Z}_{\text{chosen}}^{(t)}$ represents that particular $\boldsymbol{Z}_i, \, i = 1, 2, \ldots, n$ that is chosen for computing the sub-gradient at interation $t_s \leq t \leq t_f$. Also, we have $\tilde{\theta} \in \{\mu\theta_{t-1} + (1 - \mu) \theta_t | \, 0 \leq \mu \leq 1\}$. It is straightforward to check that due to the mentioned condition on $\alpha_t$, we have

$$\hat{R}_{\text{SSAR}} \left( \theta_{t_f}; \boldsymbol{D} \right) \leq \hat{R}_{\text{SSAR}} \left( \theta_{t_s}; \boldsymbol{D} \right). \tag{D.20}$$

On the other hand, while transitioning from the $t_f$th to $(t_f + 1)$th iteration, where at least one $y_i^* (\theta)$ changes by assumption, again we have

$$\hat{R}_{\text{SSAR}} \left( \theta_{t_f+1}; \boldsymbol{D} \right) \leq \hat{R}_{\text{SSAR}} \left( \theta_{t_f}; \boldsymbol{D} \right), \tag{D.21}$$

due to the definition of $y_i^* \left( \theta_{t_f+1} \right)$ for $i \in \mathcal{I}_{\text{ul}}$. This way, Algorithm 1 never increases the optimization objective and convergence to a stable point is guaranteed as $T \to \infty$.

Obviously, the arguments of Theorem D.1 still hold for $\delta = 0$. However, it is not practical since (E.2) cannot be solved with an infinitesimally small error in reality. On the other hand, giving a convergence rate for the two scenarios considered in this theorem, i.e. $\lambda = \pm\infty$, falls out of the scope of this paper. A trivial upper-bound on the number of iterations increases exponentially w.r.t. the

number of unlabeled samples $n_{\mathrm{ul}}$, which is based on the worst-case assumption that the combinatoric part of the optimization walks through all the possible labels for the unlabeled data. However, [11] has experimentally shown that the convergence rate (at least for a class of similar problems) is much faster. It should be noted that solving for the exact convergence rate of Theorem D.1 is equivalent to assessing the convergence rate of *self-training*, which (to the best of our knowledge) is still an open area of research. □

**Theorem D.2** (Convexity). *Assume the setting of Theorem 2 with $\Theta \subseteq \mathbb{R}^d$, for some $d \in \mathbb{N}$. Let the loss function $\ell : \mathcal{Z} \times \Theta \to \mathbb{R}_{\geq 0}$ to be twice differentiable and strictly convex with respect to $\theta$, for all $(\boldsymbol{z}, \theta) \in \mathcal{Z} \times \Theta$. Also, assume $\lambda$ satisfies the following property*

$$\lambda \geq - \inf_{(\boldsymbol{z}, \theta) \in \mathcal{Z} \times \Theta} \frac{\lambda_{\min} \left\{ \nabla_{\theta\theta}^2 \phi_\gamma \left( \boldsymbol{z}; \theta \right) \right\}}{\sigma^2 \left( 1 - |\mathcal{Y}|^{-1} \right)}, \tag{D.22}$$

*where $\nabla_{\theta\theta}^2$ is the Hessian matrix operator w.r.t. $\theta$, and $\lambda_{\min} \left\{ \cdot \right\} : \mathbb{R}^{d \times d} \to \mathbb{R}$ denotes the minimum eigenvalue operator. Then, the optimization programs in (2) and (3) w.r.t. $\theta$ are convex.*

*Proof.* For $\boldsymbol{z}_0 \in \mathcal{Z}$, let us define the function $f_{\boldsymbol{z}_0} (\theta, \boldsymbol{z}) : \Theta \times \mathcal{Z} \to \mathbb{R}$ as

$$f_{\boldsymbol{z}_0} (\theta, \boldsymbol{z}) \triangleq \ell \left( \boldsymbol{z}; \theta \right) - \gamma c \left( \boldsymbol{z}, \boldsymbol{z}_0 \right), \tag{D.23}$$

then we have $\phi_\gamma \left( \boldsymbol{z}_0; \theta \right) = \max_{\boldsymbol{z}} f_{\boldsymbol{z}_0} (\theta, \boldsymbol{z})$. Since $f$ is twice differentiable and convex w.r.t. $\theta$, $\phi_\gamma$ also shares these two properties based on Danskin's theorem [12]. Thus, the $d \times d$ hessian matrix $\nabla_{\theta\theta}^2 \phi_\gamma$ is well-defined and positive definite for all $(\boldsymbol{z}_0, \theta) \in \mathcal{Z} \times \Theta$.

By looking at (3), the first summation over labeled samples, i.e. $i \in \mathcal{I}_l$, is again a convex function w.r.t. $\theta$. However, the second summand might not be convex due to the usage of softmin. Therefore, it is sufficient to provide conditions under which $\mathrm{softmin}_{y \in \mathcal{Y}}^{(\lambda)} \left\{ \phi_\gamma \right\}$ becomes convex for all $\theta \in \Theta$. This will also prove the convexity of (3). Obviously, each softmin summand in the equation is twice differentiable and hence, for any $\boldsymbol{X} \in \mathcal{X}$, we have

$$\nabla_{\theta\theta}^2 \left( \mathop{\mathrm{softmin}}_{y \in \mathcal{Y}}^{(\lambda)} \left\{ \phi_\gamma \left( \boldsymbol{X}, y; \theta \right) \right\} \right) = \nabla_\theta \left( \sum_{y \in \mathcal{Y}} \beta_y \left( \theta \right) \nabla_\theta \phi_\gamma \left( \boldsymbol{X}, y; \theta \right) \right) \tag{D.24}$$

$$= \sum_{y \in \mathcal{Y}} \left( \beta_y \left( \theta \right) \nabla_{\theta\theta}^2 \phi_\gamma \left( \boldsymbol{X}, y; \theta \right) + \nabla_\theta \beta_y \left( \theta \right) \nabla_\theta^T \phi_\gamma \left( \boldsymbol{X}, y; \theta \right) \right),$$

where $\beta_y \left( \theta \right)$ (with $0 \leq \beta_y \left( \theta \right) \leq 1$ for $y \in \mathcal{Y}$ and $\theta \in \Theta$) is defined as

$$\beta_y \left( \theta \right) \triangleq \frac{e^{\lambda \phi_\gamma (\boldsymbol{X}, y; \theta)}}{\sum_{y' \in \mathcal{Y}} e^{\lambda \phi_\gamma (\boldsymbol{X}, y'; \theta)}} \quad , \text{and we have} \quad \sum_{y \in \mathcal{Y}} \beta_y \left( \theta \right) = 1. \tag{D.25}$$

Some mathematical simplifications reveal that

$$\nabla_\theta \beta_y \left( \theta \right) = \lambda \beta_y \left( \theta \right) \left( 1 - \beta_y \left( \theta \right) \right) \nabla_\theta \phi_\gamma \left( \boldsymbol{X}, y; \theta \right), \tag{D.26}$$

and as a result we have the following formula for the Hessian matrix of each softmin summand:

$$\nabla_{\theta\theta}^2 \left( \mathop{\mathrm{softmin}}_{y \in \mathcal{Y}}^{(\lambda)} \left\{ \phi_\gamma \left( \boldsymbol{X}, y; \theta \right) \right\} \right) = \sum_{y \in \mathcal{Y}} \beta_y \left( \theta \right) \nabla_{\theta\theta}^2 \phi_\gamma \left( \boldsymbol{X}, y; \theta \right) \tag{D.27}$$

$$+ \lambda \sum_{y \in \mathcal{Y}} \beta_y \left( \theta \right) \left( 1 - \beta_y \left( \theta \right) \right) \nabla_\theta \phi_\gamma \left( \boldsymbol{X}, y; \theta \right) \nabla_\theta^T \phi_\gamma \left( \boldsymbol{X}, y; \theta \right).$$

Note that for each $y \in \mathcal{Y}$, the $d \times d$ matrix $\nabla_\theta \phi_\gamma \left( \boldsymbol{X}, y; \theta \right) \nabla_\theta^T \phi_\gamma \left( \boldsymbol{X}, y; \theta \right)$ is rank-one, positive semi-definite and its only non-zero eigenvalue equals to $\| \nabla_\theta \phi_\gamma \left( \boldsymbol{X}, y; \theta \right) \|_2^2 \leq \sigma^2$. Therefore, the matrix corresponding to the second summand in the r.h.s. of (D.27) is negative semi-definite only if $\lambda < 0$. In this case, i.e. having a negative $\lambda$, the following upper-bound holds for the magnitude of its largest eigenvalue:

$$\leq \sigma^2 |\lambda| \max_{\boldsymbol{\beta} \in M(\mathcal{Y})} \boldsymbol{\beta}^T \left( \mathbf{1} - \boldsymbol{\beta} \right) = \sigma^2 |\lambda| \left( 1 - |\mathcal{Y}|^{-1} \right). \tag{D.28}$$

On the other hand, the first summand in the r.h.s. of (D.27) is always positive definite and (since $\beta_y(\theta)$s sum up to 1) its smallest eigenvalue satisfies the following lower-bound:

$$\geq \inf_{(\boldsymbol{z},\theta)\in\mathcal{Z}\times\Theta} \lambda_{\min}\left\{\nabla^2_{\theta\theta}\phi_\gamma(\boldsymbol{z}|\theta)\right\}. \tag{D.29}$$

Therefore, as long as i) $\lambda$ is non-negative, or ii) the upper-bound in (D.28) is strictly smaller than the lower-bound in (D.29), which is the condition of the Theorem on $\lambda$, the Hessian of $\text{softmin}^{(\lambda)}_{y\in\mathcal{Y}}\{\phi_\gamma(\boldsymbol{X},y;\theta)\}$ remains positive definite for all $\boldsymbol{z}\in\mathcal{Z}$ and $\theta\in\Theta$, and the proof is complete.

Note that due to assuming strict convexity and twice differentiability for $\ell$, $\nabla^2_{\theta\theta}\phi_\gamma$ is universally positive-definite and hence, the r.h.s. of (D.22) is negative. This argument is a direct consequence of Danskin's theorem. However, there are no general ways to directly relate eigenvalues of $\nabla^2_{\theta\theta}\ell$ to those of $\nabla^2_{\theta\theta}\phi_\gamma$, since such relations extremely depend on the properties of function $\ell$. □

*Proof of Theorem 3.* We prove the Theorem in two steps. In the first step, we show that the empirical value of the proposed semi-supervised adversarial risk, i.e. $\hat{R}_{\text{SSAR}}(\theta;\boldsymbol{D})$, *uniformly* converges to its expected value all over $\Theta$. In the second step, we use the asymptotic results of Theorem C.1 to finalize the bounds. For the first step, a similar technique to the ones used in classical learning theory, e.g. [13], is employed. In this regard, let the random variable $J(\boldsymbol{D})$ to be defined as

$$J(\boldsymbol{D}) \triangleq \sup_{\theta\in\Theta} \left|\hat{R}_{\text{SSAR}}(\theta;\boldsymbol{D}) - \mathbb{E}_{P_0}\left\{\hat{R}_{\text{SSAR}}(\theta;\boldsymbol{D})\right\}\right|. \tag{D.30}$$

On the other hand, we have $|\phi_\gamma(\boldsymbol{z};\theta)| \leq B$, for all $\boldsymbol{z}\in\mathcal{Z}$ and $\theta\in\Theta$. This can be deduced from the definition of adversarial loss $\phi_\gamma$ as follows:

$$\phi_\gamma(\boldsymbol{z};\theta) \triangleq \sup_{\boldsymbol{z}'\in\mathcal{Z}} \ell(\boldsymbol{z}';\theta) - \gamma c(\boldsymbol{z}',\boldsymbol{z}) \leq \sup_{\boldsymbol{z}'\in\mathcal{Z}} \ell(\boldsymbol{z}';\theta) \leq B,$$

$$\phi_\gamma(\boldsymbol{z};\theta) \geq \ell(\boldsymbol{z};\theta) - \gamma c(\boldsymbol{z},\boldsymbol{z}) = \ell(\boldsymbol{z};\theta) \geq -B. \tag{D.31}$$

Also, note that

$$\left|\text{softmin}^{(\lambda)}_{y\in\mathcal{Y}}\{\phi_\gamma(\boldsymbol{X},y;\theta)\}\right| \leq B, \quad \forall\lambda\in\mathbb{R}\cup\{\pm\infty\}, \tag{D.32}$$

for all $\boldsymbol{X}\in\mathcal{X}$ and $\theta\in\Theta$. Now, assume the two partially observed data sets $\boldsymbol{D}$ and $\boldsymbol{D}'$, both with size $n$, where the only difference between them is a single data point. Then, it can be readily deduced that

$$\left|\hat{R}_{\text{SSAR}}(\theta;\boldsymbol{D}) - \hat{R}_{\text{SSAR}}(\theta;\boldsymbol{D}')\right| \leq \frac{2B}{n} \quad \Rightarrow \quad |J(\boldsymbol{D}) - J(\boldsymbol{D}')| \leq \frac{2B}{n}. \tag{D.33}$$

In this regard, one can use the McDiarmid's inequality and show that: For all $0 < \delta \leq 1$, with probability at least $1 - \delta$, the following inequality holds:

$$J(\boldsymbol{D}) \leq \mathbb{E}_{P_0}\{J(\boldsymbol{D})\} + B\sqrt{\frac{2}{n}\log\frac{1}{\delta}}, \tag{D.34}$$

which also implies that the following uniform upper-bound exists for all $\theta\in\Theta$:

$$\left|\hat{R}_{\text{SSAR}}(\theta;\boldsymbol{D}) - \mathbb{E}_{P_0}\left\{\hat{R}_{\text{SSAR}}(\theta;\boldsymbol{D})\right\}\right| \leq \mathbb{E}_{P_0}\{J(\boldsymbol{D})\} + B\sqrt{\frac{2}{n}\log\frac{1}{\delta}}. \tag{D.35}$$

The term $\mathbb{E}_{P_0}\{J(\boldsymbol{D})\}$ does not depend on the randomness of the chosen dataset and is a function of the hypothesis set $\mathcal{L}$ (or more precisely, its adversarial counterpart $\Phi$), and distribution $P_0$. It plays the role of *Rademacher complexity* in classical learning theory. In order to express this term in a more intuitive formulation, first let us introduce the function $f(\boldsymbol{z},h;\theta)$ for $\boldsymbol{z} = (\boldsymbol{X},y)$ and $h\in\mathcal{H} \triangleq \{\text{l},\text{ul}\}$ as follows:

$$f(\boldsymbol{z},h;\theta) \triangleq \begin{cases} \phi_\gamma(\boldsymbol{X},y;\theta) & h = \text{l} \\ \text{softmin}^{(\lambda)}_{y\in\mathcal{Y}}\{\phi_\gamma(\boldsymbol{X},y;\theta)\} & h = \text{ul} \end{cases}, \tag{D.36}$$

where the rest of parameters are omitted from the input arguments of $f$ for the sake of simplicity in notation. It should be noted that we can write:

$$\mathbb{E}_{\boldsymbol{D}\sim P_0}\{\cdot\} = \mathbb{E}_{h_1,\ldots,h_n\in\mathcal{H}}\{\mathbb{E}_{\boldsymbol{z}_1,\ldots,\boldsymbol{z}_n\sim P_0}\{\cdot\}\} \Rightarrow \mathbb{E}_{P_0}\left\{\hat{R}_{\text{SSAR}}(\theta;\boldsymbol{D})\right\} = \mathbb{E}_h\{\mathbb{E}_{\boldsymbol{z}}\{f(\boldsymbol{z},h;\theta)\}\}$$
(D.37)

where $h_1,\ldots,h_n$ are i.i.d. bi-categorical random variables in $\mathcal{H}$, with probabilities of $\eta$ and $1-\eta$ for $h = \text{l}$ and $h = \text{ul}$, respectively. Then, Similar to [13], one can write the following set of relations:

$$\mathbb{E}_{P_0}\{J(\boldsymbol{D})\} = \mathbb{E}_{\boldsymbol{D}\sim P_0}\left\{\sup_{\theta\in\Theta}\left|\hat{R}_{\text{SSAR}}(\theta;\boldsymbol{D}) - \mathbb{E}_{\boldsymbol{D}'\sim P_0}\left\{\hat{R}_{\text{SSAR}}(\theta;\boldsymbol{D}')\right\}\right|\right\}$$

$$= \mathbb{E}_{\boldsymbol{D}\sim P_0}\left\{\sup_{\theta\in\Theta}\left|\mathbb{E}_{\boldsymbol{D}'\sim P_0}\left\{\hat{R}_{\text{SSAR}}(\theta;\boldsymbol{D}) - \hat{R}_{\text{SSAR}}(\theta;\boldsymbol{D}')\right\}\right|\right\}$$

$$\leq \mathbb{E}_{\boldsymbol{D},\boldsymbol{D}'\sim P_0}\left\{\sup_{\theta\in\Theta}\left|\hat{R}_{\text{SSAR}}(\theta;\boldsymbol{D}) - \hat{R}_{\text{SSAR}}(\theta;\boldsymbol{D}')\right|\right\}$$

$$= \mathbb{E}_{\boldsymbol{h}_{1:n},\boldsymbol{h}'_{1:n}\in\mathcal{H}}\left\{\mathbb{E}_{\boldsymbol{z}_{1:n},\boldsymbol{z}'_{1:n}\sim P_0}\left\{\sup_{\theta\in\Theta}\left|\frac{1}{n}\sum_{i=1}^n f(\boldsymbol{z}_i,h_i;\theta) - f(\boldsymbol{z}'_i,h'_i;\theta)\right|\right\}\right\}$$

$$= \mathbb{E}_{\boldsymbol{h}_{1:n},\boldsymbol{h}'_{1:n}\in\mathcal{H}}\left\{\mathbb{E}_{\boldsymbol{z}_{1:n},\boldsymbol{z}'_{1:n}\sim P_0,\,\boldsymbol{\sigma}}\left\{\sup_{\theta\in\Theta}\left|\frac{1}{n}\sum_{i=1}^n \sigma_i(f(\boldsymbol{z}_i,h_i;\theta) - f(\boldsymbol{z}'_i,h'_i;\theta))\right|\right\}\right\}$$

$$\leq 2\mathbb{E}_{\boldsymbol{h}_{1:n}\in\mathcal{H}}\left\{\mathbb{E}_{\boldsymbol{z}_{1:n}\sim P_0,\,\boldsymbol{\sigma}}\left\{\sup_{\theta\in\Theta}\left|\frac{1}{n}\sum_{i=1}^n \sigma_i f(\boldsymbol{z}_i,h_i;\theta)\right|\right\}\right\}, \tag{D.38}$$

where $\boldsymbol{\sigma}\in\{-1,+1\}^n$ represents a vector of $n$ i.i.d. Rademacher random variables. Based on this result and its preceding discussions, one can write:

$$\frac{1}{2}\mathbb{E}_{P_0}\{J(\boldsymbol{D})\} = \eta\mathbb{E}_{\boldsymbol{z}_{1:n\eta}\sim P_0,\,\boldsymbol{\sigma}}\left\{\sup_{\theta\in\Theta}\frac{1}{n\eta}\sum_{i=1}^{n\eta}\sigma_i\phi_\gamma(\boldsymbol{z}_i;\theta)\right\} \tag{D.39}$$

$$+ (1-\eta)\,\mathbb{E}_{\boldsymbol{X}_{1:n(1-\eta)}\sim P_{0\boldsymbol{X}},\,\boldsymbol{\sigma}}\left\{\sup_{\theta\in\Theta}\frac{1}{n(1-\eta)}\sum_{i=1}^{n(1-\eta)}\sigma_i\,\underset{y\in\mathcal{Y}}{\text{softmin}}^{(\lambda)}\{\phi_\gamma(\boldsymbol{X}_i,y;\theta)\}\right\}.$$

The first term in the r.h.s. of (D.39) can be more analytically investigated. In order to do so, let us define the $\epsilon$-neighborhood around $\boldsymbol{z}_0$ as $\mathcal{N}_\epsilon(\boldsymbol{z}_0) \triangleq \{\boldsymbol{z}\in\mathcal{Z}|c(\boldsymbol{z},\boldsymbol{z}_0)\leq\epsilon\}$, for $\epsilon\geq 0$. Then, there exists $\epsilon\geq 0$ such that

$$\mathbb{E}_{\boldsymbol{z}_{1:n}\sim P_0,\,\boldsymbol{\sigma}}\left\{\sup_{\theta\in\Theta}\frac{1}{n}\sum_{i=1}^n\sigma_i\phi_\gamma(\boldsymbol{z}_i;\theta)\right\} = \mathbb{E}_{\boldsymbol{z}_{1:n}\sim P_0,\,\boldsymbol{\sigma}}\left\{\sup_{\theta\in\Theta}\frac{1}{n}\sum_{i=1}^n\sigma_i\sup_{\boldsymbol{z}'_i\in\mathcal{Z}}\ell(\boldsymbol{z}'_i;\theta) - \gamma c(\boldsymbol{z}'_i,\boldsymbol{z}_i)\right\}$$

$$= \mathbb{E}_{\boldsymbol{z}_{1:n}\sim P_0,\,\boldsymbol{\sigma}}\left\{\sup_{\theta\in\Theta}\frac{1}{n}\sum_{i=1}^n\sigma_i\left[\sup_{\boldsymbol{z}'_i\in\mathcal{N}_\epsilon(\boldsymbol{z}_i)}\ell(\boldsymbol{z}'_i;\theta) - \gamma\epsilon\right]\right\}$$

$$= \mathbb{E}_{\boldsymbol{z}_{1:n}\sim P_0,\,\boldsymbol{\sigma}}\left\{\sup_{\theta\in\Theta}\frac{1}{n}\sum_{i=1}^n\sigma_i\sup_{\boldsymbol{z}'_i\in\mathcal{N}_\epsilon(\boldsymbol{z}_i)}\ell(\boldsymbol{z}'_i;\theta)\right\}$$

$$= g_{\text{l}}(n), \tag{D.40}$$

where $g_{\text{l}}(n)$ can be found in Definition 2, with the function set $\mathcal{F}$ representing the loss function set $\mathcal{L}$ in the above relations. For the second term on the r.h.s. of (D.39), the following inequality holds for all $\lambda\in\mathbb{R}\cup\{\pm\infty\}$:

$$\mathbb{E}_{\boldsymbol{X}_{1:n},\ldots,\boldsymbol{X}_n\sim P_{0\boldsymbol{X}},\,\boldsymbol{\sigma}}\left\{\sup_{\theta\in\Theta}\frac{1}{n}\sum_{i=1}^n\sigma_i\,\underset{y\in\mathcal{Y}}{\text{softmin}}^{(\lambda)}\{\phi_\gamma(\boldsymbol{X}_i,y;\theta)\}\right\}$$

$$\leq \mathbb{E}_{\boldsymbol{X}_{1:n}\sim P_{0\boldsymbol{X}},\,\boldsymbol{\sigma}}\left\{\left(\Pi_{y\in\mathcal{Y}}\sup_{\theta_y\in\Theta}\right)\frac{1}{n}\sum_{i=1}^n\sigma_i\,\underset{y\in\mathcal{Y}}{\text{softmin}}^{(\lambda)}\{\phi_\gamma(\boldsymbol{X}_i,y;\theta_y)\}\right\}$$

$$\leq \sum_{y\in\mathcal{Y}}\mathbb{E}_{\boldsymbol{z}_{1:n}\sim(P_{0\boldsymbol{X}}\delta_y),\,\boldsymbol{\sigma}}\left\{\sup_{\theta\in\Theta}\frac{1}{n}\sum_{i=1}^n\sigma_i\sup_{\boldsymbol{z}'_i\in\mathcal{N}_\epsilon(\boldsymbol{z}_i)}\ell(\boldsymbol{z}'_i;\theta)\right\} = g_{\text{ul}}(n). \tag{D.41}$$

The last two inequalities above are the results of Lemma D.6 (see below), and Definition 2, respectively. The following lemma helps us to resolve the presence of $\mathrm{softmin}$ operator in the formulation of $\mathbb{E}_{P_0}\{J(\boldsymbol{D})\}$.

**Lemma D.6.** *Assume the function sets $\mathcal{F}_j \subseteq \mathbb{R}^{\mathcal{Z}}$, $j = 1, \ldots, d$, where $\mathcal{Z}$ denotes a vector domain and $d \in \mathbb{N}$. Also, assume $\boldsymbol{\sigma} = (\sigma_1, \ldots, \sigma_d)$ to be a vector of i.i.d. Rademacher variables, and $\boldsymbol{Z} = \{\boldsymbol{z}_1, \ldots, \boldsymbol{z}_n\}$ are i.i.d. generated data points in domain $\mathcal{Z}$, according to some probability measure. Then, the following upper-bound holds for all $\lambda \in \mathbb{R} \cup \{\pm\infty\}$:*

$$\mathbb{E}_{\boldsymbol{Z},\boldsymbol{\sigma}} \left\{ \left( \prod_{j=1}^{d} \sup_{f_j \in \mathcal{F}_j} \right) \frac{1}{n} \sum_{i=1}^{n} \sigma_i \operatorname*{softmin}_{j=1,\ldots,d}^{(\lambda)} (f_j(\boldsymbol{z}_i; \theta)) \right\} \leq \sum_{j=1}^{d} \mathcal{R}_n(\mathcal{F}_j), \qquad \text{(D.42)}$$

*where $\mathcal{R}_n(\cdot)$ denotes the $n$-point expected Rademacher complexity w.r.t. to the same distribution that generates the samples in $\boldsymbol{Z}$.*

*Proof.* Looking at the definition of $\mathrm{softmin}$ in (4), first let us consider the following function: For $a, b \in \mathbb{R}$ and non-negative parameters $\alpha$ and $\beta$, with $\alpha + \beta = 1$, define

$$H_{\lambda,\alpha,\beta}(a, b) \triangleq \frac{1}{\lambda} \log\left( \alpha e^{\lambda a} + \beta e^{\lambda b} \right). \qquad \text{(D.43)}$$

Then, the following relations hold:

$$H_{\lambda,\alpha,\beta}(a, b) = a + \frac{1}{\lambda} \log\left( \alpha + \beta e^{\lambda(b-a)} \right)$$

$$= b + \frac{1}{\lambda} \log\left( \beta + \alpha e^{\lambda(a-b)} \right) \qquad \text{(D.44)}$$

and, as a result

$$H_{\lambda,\alpha,\beta}(a, b) = \frac{a}{2} + \frac{b}{2} + \frac{1}{2\lambda} \left[ \log\left( \alpha + \beta e^{\lambda(b-a)} \right) + \log\left( \beta + \alpha e^{\lambda(a-b)} \right) \right]$$

$$\triangleq \frac{a+b}{2} + h_{\lambda,\alpha,\beta}(b-a), \qquad \text{(D.45)}$$

where the last equality is in fact the definition of $h_{\lambda,\alpha,\beta} : \mathbb{R} \to \mathbb{R}$. It should be noted that $h_{\lambda,\alpha,\beta}(0) = 0$. Also, the following holds for the derivative of $h_{\lambda,\alpha,\beta}(\cdot)$:

$$h'_{\lambda,\alpha,\beta}(u) = \frac{\beta^2 e^{\lambda u} - \alpha^2 e^{-\lambda u}}{2\alpha\beta + \beta^2 e^{\lambda u} + \alpha^2 e^{-\lambda u}} = \frac{\beta e^{(\lambda u)/2} - \alpha e^{(-\lambda u)/2}}{\beta e^{(\lambda u)/2} + \alpha e^{(-\lambda u)/2}}, \qquad \text{(D.46)}$$

which indicates $\left| h'_{\lambda,\alpha,\beta}(u) \right| \leq 1$, for all $u \in \mathbb{R}$ and the legitimate set of parameters $(\lambda, \alpha, \beta)$. Therefore, $h'_{\lambda,\alpha,\beta}$ is a 1-Lipschitz continuous function. In this regard, for any two real-valued function sets $\mathcal{A}$ and $\mathcal{B}$ whose domain is $\mathcal{Z}$, the following relation holds due to the *sum inequality* of Rademacher complexity:

$$\mathbb{E}_{\boldsymbol{Z},\boldsymbol{\sigma}} \left\{ \sup_{a \in \mathcal{A}, \, b \in \mathcal{B}} \frac{1}{n} \sum_{i=1}^{n} \sigma_i H_{\lambda,\alpha,\beta}(a(\boldsymbol{z}_i), b(\boldsymbol{z}_i)) \right\} = \mathcal{R}_n \left( \left\{ \frac{a+b}{2} + \frac{1}{2} h_{\lambda,\alpha,\beta}(b-a) \, \middle| \, a \in \mathcal{A}, \, b \in \mathcal{B} \right\} \right)$$

$$\leq \frac{1}{2} \left[ \mathcal{R}_n(\mathcal{A}) + \mathcal{R}_n(\mathcal{B}) + \mathcal{R}_n(h_{\lambda,\alpha,\beta} \circ \mathcal{C}) \right], \qquad \text{(D.47)}$$

where $\mathcal{C} \triangleq \{ a - b \, | \, a \in \mathcal{A}, \, b \in \mathcal{B} \}$. It can be readily verified that $\mathcal{R}_n(\mathcal{C}) \leq \mathcal{R}_n(\mathcal{A}) + \mathcal{R}_n(\mathcal{B})$. Also, *Talagrand's contraction lemma* in statistical learning theory [13] states that given the above properties for a 1-Lipschitz function $h_{\lambda,\alpha,\beta}(\cdot)$, we have $\mathcal{R}_n(h_{\lambda,\alpha,\beta} \circ \mathcal{C}) \leq \mathcal{R}_n(\mathcal{C})$. Therefore, the previous chain of inequalities can be concluded as

$$\mathcal{R}_n \left( H_{\lambda,\alpha,\beta}(a, b) \, \middle| \, a \in \mathcal{A}, b \in \mathcal{B} \right) \leq \mathcal{R}_n(\mathcal{A}) + \mathcal{R}_n(\mathcal{B}). \qquad \text{(D.48)}$$

for all $\lambda \in \mathbb{R} \cup \{\pm\infty\}$, and all $\alpha, \beta \geq 0$ with $\alpha + \beta = 1$. For the remainder of the proof, one should consider the following recursive relation for all $f_1, \ldots, f_d$:

$$\operatorname*{softmin}_{j=1,\ldots,d}^{(\lambda)} (f_j) = H_{\lambda, \frac{d-1}{d}, \frac{1}{d}} \left( \operatorname*{softmin}_{j=1,\ldots,d-1}^{(\lambda)} (f_j), f_d \right), \qquad \text{(D.49)}$$

which can be verified through a simple substitution of parameters. By using (D.48), we have

$$\mathcal{R}_n \left( \underset{j=1,\ldots,d}{\overset{(\lambda)}{\mathrm{softmin}}} (f_j) \,\Big|\, f_j \in \mathcal{F}_j \right) \leq \mathcal{R}_n \left( \underset{j=1,\ldots,d-1}{\overset{(\lambda)}{\mathrm{softmin}}} (f_j) \,\Big|\, f_j \in \mathcal{F}_j \right) + \mathcal{R}_n \left( \mathcal{F}_d \right). \qquad \text{(D.50)}$$

Repeating the above inequality for $d$ consecutive times gives us the desired result and completes the proof. $\qquad\square$

According to Definition 2, the previous upper-bounds can be simplified into the following statement: With probability at least $1 - \delta$, and for all $\theta \in \Theta$, we have

$$\left| \hat{R}_{\mathrm{SSAR}} \left( \theta; \boldsymbol{D} \right) - \mathbb{E}_{P_0} \left\{ \hat{R}_{\mathrm{SSAR}} \left( \theta; \boldsymbol{D} \right) \right\} \right| \leq 2 \left( \mathcal{R}_{n,(\epsilon,\eta)}^{(\mathrm{SSM})} \left( \mathcal{L} \right) + B \sqrt{\frac{\log \frac{1}{\delta}}{2n}} \right), \qquad \text{(D.51)}$$

where $\epsilon \geq 0$ is the dual counterpart of $\gamma$ in (3). Therefore, the empirical values of $R_{\mathrm{SSAR}}$ are always close (and asymptotically convergent) to their corresponding expected values. Next, we have to show that the expected value of $R_{\mathrm{SSAR}}$ legitimately upper-bounds the true risk at the solution point, i.e. $\theta^* \in \Theta$.

Let $\theta_{\mathrm{true}}^*$ to represent the true minimizer of the expected adversarial risk, i.e. $\theta_{\mathrm{true}}^* \triangleq \mathrm{argmin}_{\theta \in \Theta} \mathbb{E}_{P_0} \left\{ \phi_\gamma \left( \boldsymbol{Z}; \theta \right) \right\}$. Then, based on Theorem C.1 and for any $\zeta \geq 0$, there exists a neighborhood around $\theta_{\mathrm{true}}^*$, denoted by $\Theta_{\mathrm{local}} \subset \Theta$, such that the following gap is guaranteed to exist for all $\theta \notin \Theta_{\mathrm{local}}$:

$$\mathbb{E}_{P_0} \left\{ \hat{R}_{\mathrm{SSAR}} \left( \theta; \boldsymbol{D} \right) - \hat{R}_{\mathrm{SSAR}} \left( \theta_{\mathrm{true}}^*; \boldsymbol{D} \right) \right\} \geq \zeta, \qquad \text{(D.52)}$$

given that the condition $\eta \geq \mathrm{MSR}_{(\Phi, P_0)} \left( \lambda, \zeta \right)$ is satisfied. According to the assumption on $\eta$ in the current theorem, it can be readily deduced that with probability at least $1 - \delta$, the following relation holds for all $\theta \notin \Theta_{\mathrm{local}}$:

$$\hat{R}_{\mathrm{SSAR}} \left( \theta; \boldsymbol{D} \right) - \hat{R}_{\mathrm{SSAR}} \left( \theta_{\mathrm{true}}^*; \boldsymbol{D} \right) > 0 \quad \Rightarrow \quad \theta^* \triangleq \underset{\theta \in \Theta}{\mathrm{argmin}} \, \hat{R}_{\mathrm{SSAR}} \left( \theta; \boldsymbol{D} \right) \in \Theta_{\mathrm{local}}, \qquad \text{(D.53)}$$

i.e. the minimizer of $\hat{R}_{\mathrm{SSAR}} \left( \theta; \boldsymbol{D} \right)$ also falls in $\Theta_{\mathrm{local}}$. Also, for all $\theta \in \Theta_{\mathrm{local}}$ and any $\epsilon \geq 0$ we have

$$\mathbb{E}_{P_0} \left\{ \hat{R}_{\mathrm{SSAR}} \left( \theta; \boldsymbol{D} \right) \right\} \geq \mathbb{E}_{P_0} \left\{ \phi_\gamma \left( \boldsymbol{Z}; \theta \right) \right\} + \gamma \epsilon \geq \underset{P \in \mathcal{B}_\epsilon (P_0)}{\mathrm{sup}} \mathbb{E}_P \left\{ \ell \left( \boldsymbol{Z}; \theta \right) \right\}. \qquad \text{(D.54)}$$

Combining relations given in (D.51), (D.53) and (D.54) gives the desired result and completes the proof. $\qquad\square$

# E  Auxiliary Lemmas and Proofs

**Lemma E.1.** *Consider the setting described in Theorem 1. Assume $\ell \left( \boldsymbol{z}; \theta \right)$ is differentiable w.r.t. $\boldsymbol{z}$, and $\nabla_{\boldsymbol{z}} \ell \left( \cdot; \theta \right)$ is $L_{zz}$-Lipschitz all over $\mathcal{Z} \times \Theta$, for some $L_{zz} \geq 0$. Also, assume transportation cost $c$ is $1$-strongly convex in its first argument. Then, if $\gamma > L_{zz}$, the program*

$$\underset{\boldsymbol{z}' \in \mathcal{Z}}{\mathrm{sup}} \, \ell \left( \boldsymbol{z}'; \theta \right) - \gamma c \left( \boldsymbol{z}', \left( \boldsymbol{X}, y \right) \right) \qquad \text{(E.1)}$$

*becomes $(\gamma - L_{zz})$-strongly concave for all $\left( \boldsymbol{X}, y \right) \in \mathcal{Z}$.*

The proof is straightforward and uses Taylor's expansion series. Actually, it directly results from the definition of $\gamma$-concavity.

**Lemma E.2.** *Assume loss function $\ell : \mathcal{Z} \times \Theta \to \mathbb{R}$, $c : \mathcal{Z} \times \mathcal{Z} \to \mathbb{R}_{\geq 0}$ and $\gamma \geq 0$, such that conditions in Lemma E.1 hold all over $\mathcal{Z} \times \Theta$. Assume $\ell$ is differentiable w.r.t. $\theta$, and let $\boldsymbol{g}_\theta \left( \boldsymbol{z} \right) \triangleq \nabla_\theta \ell \left( \boldsymbol{z}; \theta \right)$. For a fixed $\theta \in \Theta$ and $i \in \mathcal{I}_{\mathrm{l}}$, define $\boldsymbol{z}_i^* \left( \theta \right)$ as the maximizer of (E.1) for $\left( \boldsymbol{X}_i, y_i \right)$. Similarly, let $\boldsymbol{z}_i^* \left( y; \theta \right)$ to represent the maximizer of*

$$J_i \left( y; \theta \right) \triangleq \underset{\boldsymbol{z}' \in \mathcal{Z}}{\mathrm{sup}} \, \ell \left( \boldsymbol{z}'; \theta \right) - \gamma c \left( \boldsymbol{z}', \left( \boldsymbol{X}_i, y \right) \right), \quad y \in \mathcal{Y}, i \in \mathcal{I}_{\mathrm{ul}}. \qquad \text{(E.2)}$$

*Then, the gradient of* (3) *w.r.t.* $\theta \in \Theta$ *can be attained as*

$$\nabla_\theta \hat{R}_{\text{SSAR}}(\theta; \boldsymbol{D}) = \frac{1}{n} \sum_{i \in \mathcal{I}_1} \boldsymbol{g}_\theta(\boldsymbol{z}_i^*(\theta)) + \frac{1}{n} \sum_{i \in \mathcal{I}_{\text{ul}}} \sum_{y \in \mathcal{Y}} q(y; \theta) \boldsymbol{g}_\theta(\boldsymbol{z}_i^*(y; \theta)), \qquad \text{(E.3)}$$

*where* $q(y; \theta) \triangleq \exp(\lambda J_i(y; \theta)) / \left( \sum_{y' \in \mathcal{Y}} \exp(\lambda J_i(y'; \theta)) \right).$

Proof of Lemma E.2 is included in that of Theorem 2, which is in Appendix D.

*Proof of Lemma D.4.* For simplicity, let us consider the following change of notation: for a fixed $\boldsymbol{X} \in \mathcal{X}$ and $\lambda \in \mathbb{R}$, define:

$$f(\theta) \triangleq \underset{y \in \mathcal{Y}}{\overset{(\lambda)}{\text{softmin}}} \{\phi_\gamma(\boldsymbol{X}, y; \theta)\}, \qquad \text{(E.4)}$$

where $\boldsymbol{X}$ and $\lambda$ are hidden from $f$. Then, based on the definition of softmin, it can be easily verified that we have the following formulation for $\nabla_\theta f$:

$$\nabla_\theta f = \sum_{y \in \mathcal{Y}} \beta_y(\theta) \nabla_\theta \phi_\gamma(\boldsymbol{X}, y; \theta) \quad \text{with} \quad \beta_y(\theta) \triangleq \frac{e^{\lambda \phi_\gamma(\boldsymbol{X}, y; \theta)}}{\sum_{y' \in \mathcal{Y}} e^{\lambda \phi_\gamma(\boldsymbol{X}, y'; \theta)}}, \ y \in \mathcal{Y}, \qquad \text{(E.5)}$$

where $\sum_{y \in \mathcal{Y}} \beta_y(\theta) = 1$, for all $\theta \in \Theta$. Hence, the following inequalities hold:

$$\begin{aligned}
\|\nabla_\theta f(\theta) - \nabla_\theta f(\theta')\|_* &= \left\| \sum_{y \in \mathcal{Y}} \beta_y(\theta) \nabla_\theta \phi_\gamma(\boldsymbol{X}, y; \theta) - \sum_{y \in \mathcal{Y}} \beta_y(\theta') \nabla_\theta \phi_\gamma(\boldsymbol{X}, y; \theta') \right\|_* \\
&\leq \sum_{y \in \mathcal{Y}} \beta_y(\theta) \|\nabla_\theta \phi_\gamma(\boldsymbol{X}, y; \theta) - \nabla_\theta \phi_\gamma(\boldsymbol{X}, y; \theta')\|_* \\
&\quad + \sum_{y \in \mathcal{Y}} \|\nabla_\theta \phi_\gamma(\boldsymbol{X}, y; \theta')\|_* |\beta_y(\theta) - \beta_y(\theta')| \\
&\leq \sum_{y \in \mathcal{Y}} \beta_y(\theta) \left( L_{\theta\theta} + \frac{L_{\boldsymbol{z}\theta} L_{\theta \boldsymbol{z}}}{\gamma - L_{\boldsymbol{z}\boldsymbol{z}}} \right) \|\theta - \theta'\| + \sigma \omega |\mathcal{Y}| \|\theta - \theta'\| \\
&= \left( L_{\theta\theta} + \frac{L_{\boldsymbol{z}\theta} L_{\theta \boldsymbol{z}}}{\gamma - L_{\boldsymbol{z}\boldsymbol{z}}} + \sigma \omega |\mathcal{Y}| \right) \|\theta - \theta'\|, \qquad \text{(E.6)}
\end{aligned}$$

where $\omega$ denotes the Lipschitz constant of $\beta_y(\theta)$ w.r.t. $\theta$, for all $y \in \mathcal{Y}$. The last inequality is a direct consequence of assuming $\|\nabla_\theta \phi_\gamma(\boldsymbol{X}, y; \theta)\|_* \leq \sigma$, which can be validated through the following mathematical argument: There exists $\epsilon \geq 0$, such that

$$\begin{aligned}
\|\nabla_\theta \phi_\gamma(\boldsymbol{X}, y; \theta)\|_* &= \left\| \nabla_\theta \left( \sup_{\boldsymbol{z}' \in \mathcal{Z}} \ell(\boldsymbol{X}, y; \theta) - \gamma c(\boldsymbol{z}', (\boldsymbol{X}, y)) \right) \right\|_* \\
&= \left\| \nabla_\theta \ell \left( \underset{\boldsymbol{z}' \in \mathcal{Z}}{\text{argmax}} \ \ell(\boldsymbol{z}'; \theta) - \gamma c(\boldsymbol{z}', (\boldsymbol{X}, y)); \theta \right) \right\|_* \leq \sigma, \qquad \text{(E.7)}
\end{aligned}$$

where the last inequality is due to the assumption of Lemma D.3 under an appropriate choice of norm. The middle equality in (E.7) is the result of the extended Danskin's theorem which relaxes convexity into *inf-compactness* of function $\ell$. For proof of inf-compactness of $\ell$ and the consequent properties, see Section 4 of [12]. In order to assess $\omega$, which is an indicator of smoothness for $\beta_y(\theta)$, one can take advantage of the *Mean Value Theorem* [14], as follows:

$$|\beta_y(\theta) - \beta_y(\theta')| \leq \max_{y \in \mathcal{Y}} \sup_{\theta^* \in \mathcal{T}(\theta \to \theta')} \|\nabla_\theta \beta_y(\theta^*)\| \|\theta - \theta'\|, \quad \theta, \theta' \in \Theta, \qquad \text{(E.8)}$$

where $\mathcal{T}(\theta \to \theta')$ is the set of all continuous paths from $\theta$ to $\theta'$ that entirely lie in $\Theta$. It is not hard to verify that the gradient $\nabla_\theta \beta_y(\theta)$ has the following formulation:

$$\nabla_\theta \beta_y(\theta) = \lambda \beta_y(\theta) \sum_{y' \in \mathcal{Y}} \beta_{y'}(\theta) \left( \nabla_\theta \phi_\gamma(\boldsymbol{X}, y; \theta) - \nabla_\theta \phi_\gamma(\boldsymbol{X}, y'; \theta) \right), \quad (y, \theta) \in \mathcal{Y} \times \Theta, \quad \text{(E.9)}$$

and hence satisfies the subsequent inequalities:

$$\|\nabla_\theta \beta_y(\theta)\| \leq 2|\lambda| \sum_{y' \in \mathcal{Y}} \beta_{y'}(\theta) \max_{h \in \{y,y'\}} \{\|\nabla_\theta \phi_\gamma(\boldsymbol{X}, h|\theta)\|\} \leq 2\sigma|\lambda|, \quad \forall \theta \in \Theta. \qquad (\text{E.10})$$

Combining (E.8) with (E.10) provides us with the safe choice of $\omega = 2\sigma|\lambda|$. Therefore, $\nabla_\theta f$ is $\left(L_{\theta\theta} + \frac{L_{z\theta}L_{\theta z}}{\gamma - L_{zz}} + 2\sigma^2|\lambda||\mathcal{Y}|\right)$-Lipschitz w.r.t. $\theta$, and the proof is complete. $\qquad \square$

*Proof of Lemma D.5.* The proof is simple and directly results from the assumptions. According to the differentiablility of $\phi_\gamma$ w.r.t. $\theta$ which is a consequence of an extended version of Danskin's theorem (see Lemma D.4), the following relations hold:

$$\|\nabla_\theta \phi_\gamma(\boldsymbol{z}_0; \theta) - \nabla_\theta \ell(\hat{\boldsymbol{z}}^*; \theta)\|_* = \left\| \nabla_\theta \ell\left( \operatorname*{argmax}_{\boldsymbol{z}' \in \mathcal{Z}} \ell(\boldsymbol{z}'; \theta) - \gamma c(\boldsymbol{z}', \boldsymbol{z}_0); \theta \right) - \nabla_\theta \ell(\hat{\boldsymbol{z}}^*; \theta) \right\|_*$$

$$\leq L_{\theta z} \left\| \hat{\boldsymbol{z}}^* - \operatorname*{argmax}_{\boldsymbol{z}' \in \mathcal{Z}} (\ell(\boldsymbol{z}'; \theta) - \gamma c(\boldsymbol{z}', \boldsymbol{z}_0)) \right\|. \qquad (\text{E.11})$$

On the other hand, due to $(\gamma - L_{zz})$-strict-concavity of (E.2), a $\delta$-approximation maximizer, i.e. $\hat{\boldsymbol{z}}^*$, satisfies

$$\left\| \hat{\boldsymbol{z}}^* - \operatorname*{argmax}_{\boldsymbol{z}' \in \mathcal{Z}} (\ell(\boldsymbol{z}'; \theta) - \gamma c(\boldsymbol{z}', \boldsymbol{z}_0)) \right\|^2 \leq \frac{L_{z\theta}}{L_{\theta z}(\gamma - L_{zz})}. \qquad (\text{E.12})$$

Substituting the above into (E.11) completes the proof. $\qquad \square$

**Lemma E.3.** *Assume a feature-label space $\mathcal{Z} = \mathcal{X} \times \mathcal{Y}$ and a function class $\mathcal{F} \subseteq \mathbb{R}^{\mathcal{Z}}$, for a feature space $\mathcal{X}$ and a finite label set $\mathcal{Y}$. Also, assume there exists $\Delta : \mathbb{N} \to \mathbb{R}$, such that $\mathcal{R}_n(\mathcal{F}) \leq \Delta(n)$, for all $n \in \mathbb{N}$ and any data distribution $P_0 \in M(\mathcal{Z})$. Then, the following holds:*

$$\mathcal{R}_{n,(\epsilon,\eta)}^{(\text{SSM})}(\mathcal{F}) \leq \eta \Delta(\lceil \eta n \rceil) + (1-\eta)|\mathcal{Y}|\Delta(\lceil (1-\eta)n \rceil) \qquad (\text{E.13})$$

*for all distributions in $M(\mathcal{Z})$, any $\epsilon \geq 0$ and $\eta \in [0,1]$.*

*Proof.* According to the assumption, $\Delta(n)$ is an upper-bound for Rademacher complexity of $\mathcal{F}$, regardless of the probability measure that generates the data samples. Therefore, one can write

$$\sup_{P_0 \in M(\mathcal{Z})} \mathbb{E}_{\boldsymbol{z}_{1:n} \sim P_0, \boldsymbol{\sigma}} \left\{ \sup_{f \in \mathcal{F}} \frac{1}{n} \sum_{i=1}^n \sigma_i f(\boldsymbol{z}_i) \right\} = \sup_{\boldsymbol{z}_{1:n} \in \mathcal{Z}} \mathbb{E}_{\boldsymbol{\sigma}} \left\{ \sup_{f \in \mathcal{F}} \frac{1}{n} \sum_{i=1}^n \sigma_i f(\boldsymbol{z}_i) \right\} \leq \Delta(n). \qquad (\text{E.14})$$

In this regard, the following relations hold for the function $g_l(n)$ of Definition 2:

$$g_l(n) = \mathbb{E}_{\boldsymbol{z}_{1:n} \sim P_0, \boldsymbol{\sigma}} \left\{ \sup_{f \in \mathcal{F}} \frac{1}{n} \sum_{i=1}^n \sigma_i \left[ \sup_{a \in \mathcal{A}_\epsilon} f(a(\boldsymbol{z}_i)) \right] \right\}$$

$$\leq \mathbb{E}_{\boldsymbol{z}_{1:n} \sim P_0, \boldsymbol{\sigma}} \left\{ \sup_{\boldsymbol{z}'_{1:n} \in \mathcal{Z} | c(\boldsymbol{z}_i, \boldsymbol{z}'_i) \leq \epsilon} \sup_{f \in \mathcal{F}} \frac{1}{n} \sum_{i=1}^n \sigma_i [f(\boldsymbol{z}'_i)] \right\}$$

$$\leq \sup_{\boldsymbol{z}'_{1:n} \in \mathcal{Z}} \mathbb{E}_{\boldsymbol{\sigma}} \left\{ \sup_{f \in \mathcal{F}} \frac{1}{n} \sum_{i=1}^n \sigma_i f(\boldsymbol{z}'_i) \right\} \leq \Delta(n). \qquad (\text{E.15})$$

With some very similar mathematical arguments, one can easily show that $g_{ul}(n) \leq |\mathcal{Y}|\Delta(n)$. Therefore, for any distribution $P_0$, any $\epsilon \geq 0$ and any $\eta \in [0,1]$, we always have

$$\mathcal{R}_{n,(\epsilon,\eta)}^{(\text{SSM})} \triangleq \eta g_l(\lceil n\eta \rceil) + (1-\eta) g_{ul}(\lceil n(1-\eta) \rceil)$$

$$\leq \Delta(\lceil n\eta \rceil) + (1-\eta)|\mathcal{Y}|\Delta(\lceil n(1-\eta) \rceil). \qquad (\text{E.16})$$

and the proof is complete.

In particular, assume a 0-1 loss function set $\mathcal{L} = \{\ell\left(\cdot;\theta\right) \mid \theta \in \Theta\}$, where $\Theta$ denotes the parameter space of a classifier with a finite VC-dimension of $\dim\left(\Theta\right)$. Then, due to Dudley's entropy bound and Haussler's upper-bound [13], there exists constant $C$ such that

$$\Delta\left(n\right) = C\sqrt{\frac{\dim\left(\Theta\right)}{n}} \tag{E.17}$$

is a valid upper-bound on the Rademacher complexity of $\mathcal{F}$ regardless of $P_0$. Then, one can write

$$
\begin{aligned}
\mathcal{R}_{n,(\epsilon,\eta)}^{(\mathrm{SSM})} &\leq \Delta\left(\lceil n\eta \rceil\right) + \left(1 - \eta\right)\left|\mathcal{Y}\right|\Delta\left(\lceil n\left(1-\eta\right)\rceil\right) \\
&= C\left[\eta\sqrt{\frac{\dim\left(\Theta\right)}{\lceil n\eta \rceil}} + \left(1-\eta\right)\left|\mathcal{Y}\right|\sqrt{\frac{\dim\left(\Theta\right)}{\lceil n\left(1-\eta\right)\rceil}}\right] \\
&\leq C\left[\eta\sqrt{\frac{\dim\left(\Theta\right)}{n\eta}} + \left(1-\eta\right)\left|\mathcal{Y}\right|\sqrt{\frac{\dim\left(\Theta\right)}{n\left(1-\eta\right)}}\right] \\
&= C\sqrt{\frac{\dim\left(\Theta\right)}{n}}\left[\sqrt{\eta} + \left|\mathcal{Y}\right|\sqrt{1-\eta}\right].
\end{aligned}
\tag{E.18}
$$

This will also prove the claim on SSM Rademacher complexity in Section 2.2. $\qquad\square$