[Reviews · NeurIPS 2019]

Reviewer 1



Update after reading the rebuttal and discussion with other reviewers: I feel my background knowledge is not enough for giving a strong support for this submission, therefore I decreased my score. ------- I should admit that I am not familiar with this area. Originality: Good. I think the notion of minimum supervision ratio is interesting and novel. Moreover, they can use a hyperparameter lambda to denote control the quality of the unlabeled data. And their results subsumes the cluster assumption, which is widely used by semi-supervised learning papers. Quality: Good. I did not check the proofs in the appendix. But the statements look reasonable to me, and the logic flow is clear and convincing. The authors clearly discussed existing results as special cases of their result, e.g. the cluster assumption. Clarify: Good. This paper is nicely presented and structured. Significance: Good. Semi-supervised learning is widely used in practice, and it is important to have theoretical understanding of the algorithms and sample complexity. Therefore I think this paper is a nice contribution to this topic, and many people in the NeurIPS community will be interested.

Reviewer 2



This paper proposes a theoretical framework which combines Semi-Supervised Learning (SSL) and Distributionally Robust Learning (DRL). The authors added an entropy regularizer on the unlabeled data, and analyzes the generalization error of the solution to an Lagrangian relaxation. The generalization bound is based on two new complexity measures, semi-supervised monge (SSM) complexity and minimun supervision ratio. The authors showed the connection of SSM and VC dimension, which is nice. The proposed algorithms hows a comparable performance to those of the state-of-the-art on a number of real-world benchmark datasets. Major Comments: It is unclear why the proposed SSM measure can explain the help from the unlabeled data. Consider a purely supervised algorithm using only n*eta labeled data from the dataset and just optimizes the supervised part of the loss. We can use the standard Rademacher complexity to derive a generalization bound. How does this bound compare to the SSM-based generalization bound? Under what condition does unlabeled data help generalization? Minor Comments: 1. It seems possible to define SSM complexity without the distribution robustness (so it can be applied to general semi-supervised learning). Is there any existing work about this? 2. The minimum supervision ratio has a very implicit dependence on lambda and zeta, so I am not sure about how useful is this complexity measure. Is there a way to know MSR empirically?

Reviewer 3



The paper is well written and is easy to follow. The proposed methodology seems to work well.

Reviewer 4



** I noticed that one of the files in the codes-for-reproducibility, submitted by the authors, contains a name-like-string which could possibly be of authors (see dmuxspec.yml). This happened after having formed an opinion about the submission, fortunately, but I am not sure if this is a violation of reviewer's code of conduct (or equivalent).** I believe that the submission could be something much more than a 'self-learning version' of Sinha, Namkoong and Duchi [9], if the authors put additional efforts in persuading the readers of the importance of the proposed notions, e.g. minimum supervision ratio and SSM Rademacher complexity. For the readers who are already familiar of the results in [9] or [11], I do not think Theorem 1, 2, or 3 *per se* would come as surprising or significant; the proof ideas are already quite well-established. Instead, I will be more interested in the nature of newly suggested notions, such as MSR, and an in-depth analysis on the objects would be a more contributing fraction of the submission. For example, it would be quite interesting to see if MSR could be characterized explicitly under simple scenarios, from which one could understand the dependencies of MSR to various parameters. Here are a few minor remarks: 1. I noticed that in section 1.2, authors use distributional robustness learning quite interchangeably with adversarially robustness, which are quite different subjects to me, in the sense that the latter is more about pointwise perturbations instead of distributional. 2. Is n equal to n_l + n_{ul}? This could cause unnecessary confusion, as in Section 1.1. n is introduced as a cardinality of the dataset D = {Z_1, ... Z_n}, without any context of being semi-supervised. 3. Missing a space between sentences, at line 222. 4. Typo at line 246, denot -> denote. 5. Some readers may be interested in seeing the choice of model in the main text, to be more critical about the robustness guaranteeable by semi-supervised methods. 6. The title of the reference [20] should be "Monge blunts Bayes:...," if I am correct.

[Author Response · NeurIPS 2019]

We would like to thank all the four reviewers for their comments. In this document, we try to briefly respond to the concerns and questions raised by Reviewers 2 and 4.

**Reviewer 2**: –(**major comments**) We have tried to avoid making any claims about "explaining the help from unlabeled data". Theorem 3 only provides a generalization bound. We revise the paper again to remove any remaining claims about this issue. However, using our framework, one can (at least theoretically) characterize cases where unlabeled data can provably help. First, it should be noted that by using only the labeled data for learning (as suggested by the reviewer), the residual generalization error in the classical learning framework would be $O\left(n^{-1/2}\eta^{-1/2}\right)$. But residual error terms of Theorem 3 are both $O\left(n^{-1/2}\right)$ (note that $\sqrt{\eta} + \sqrt{1-\eta} \leq \sqrt{2}$). Therefore, we can guarantee a much smaller residual error when supervision ratio is very small, i.e. $\eta \ll 1$. Second, for a highly compatible pair of model set $\mathbf{\Phi}$ and data distribution $P_0$, the condition $\mathrm{MSR}_{\mathbf{\Phi},P_0}\left(\lambda, \zeta\right) < \eta$ can be satisfied even for very small (and generally negative) values of $\lambda$. For a sufficiently small $\lambda$, our $\hat{R}_{\mathrm{SSAR}}$ becomes smaller than the average risk computed over only the labeled data. Let us discuss this matter, mathematically: For simplicity, assume the asymptotic case of $n \to +\infty$ (similar arguments hold for $n < +\infty$). Then, with a little abuse of notation and for any $\phi \in \mathbf{\Phi}$, we have:

$$\lim_{n\to+\infty} \hat{R}_{\mathrm{SSAR}}\left(\phi; \boldsymbol{D}\right) \overset{a.s.}{=} \mathbb{E}_{\boldsymbol{X}\sim P_{0_{\boldsymbol{X}}}}\left\{\eta\mathbb{E}_{y\sim P_{0_{|\boldsymbol{X}}}}\left\{\phi\left(\boldsymbol{X},y\right)\right\} + (1-\eta)\overset{(\lambda)}{\underset{y\in\mathcal{Y}}{\mathrm{softmin}}}\left\{\phi\left(\boldsymbol{X},y\right)\right\}\right\} \overset{*}{\leq} \mathbb{E}_{\boldsymbol{X},y\sim P_0}\left\{\phi\left(\boldsymbol{X},y\right)\right\},$$

where $*$ holds for sufficiently small values of $\lambda$, since $\mathbb{E}_{y\sim P_{0_{|\boldsymbol{X}}}}$ is an expectation operator but $\mathrm{softmin}^{(\lambda)}$ can go as far as being the $\min$ operator. Therefore, one can establish a set of theoretical conditions under which unlabeled data is guaranteed to be helpful, since all the three terms in the r.h.s. of the bound in Theorem 3 become smaller than their traditional counterparts. The above-mentioned conditions are very general, but at the same time very implicit. In any case, we will add a lemma to our appendix to highlight this issue for interested readers.

–(**minor comments**) 1. Yes, our SSM measure can also be used when $\epsilon = 0$ (i.e. no distributional robustness). To the best of our knowledge, there are no similar theoretical treatments of this problem in the existing works. 2. Please refer to our response to Reviewer 4.

**Reviewer 4**:–(**major comments**) Considering reviewer's comments, first let us emphasize on some of our contributions that might have been missed during the review: we have tested our method on three different datasets and outperformed state-of-the-art in at least one of them. Also, we theoretically showed that a model set with a bounded VC-dimension is also *adversarially-learnable* (Lemma E.3), even in a *semi-supervised* scenario, where a corresponding generalization bound is given in Theorem 3. We agree with both Reviewers 2 and 4 that MSR in (C.11) is very implicit and hard to evaluate. However, please note that our framework is completely general, and thus providing a way to evaluate MSR in a general scenario might lead to solving several open problems in statistics (similar to providing a general way to evaluate VC-dimension or Rademacher complexity for any model set). For example, consider the loss function set $\mathbf{\Phi} = \{-\log P_\theta\left(\cdot,\cdot\right) \mid \theta \in \Theta\}$, where $P_\theta$ can be any parametric distribution family over $\mathcal{X} \times \mathcal{Y}$. Also, assume dataset is sampled from $P_{\theta_0}$, where $\theta_0 \in \Theta$. Then, it can be easily seen that the proposed risk in Theorem 1 when $\lambda = -1$, is in fact the ML estimator (which is also the optimal estimator). Characterizing MSR in this case can shed light on the sample complexity of ML in a general semi-supervised setting which is still an open problem. However, let us give a quick example of how fast MSR can be computed in some very specific and simple cases: Assume the *cluster assumption*, where data distribution $P_0$ is a mixture of two distributions whose supports do not overlap over $\mathcal{X}$, and correspond to only $y = -1$ and $+1$ over $\mathcal{Y}$, respectively. Consider the loss function set $\mathbf{\Phi}$ which is associated with a family of arbitrary binary classifiers, where for each $\phi \in \mathbf{\Phi}$ we have $\phi\left(\boldsymbol{X},y\right) = \infty \cdot \phi_{acc}\left(\boldsymbol{X},y\right) + \phi_{mar}\left(\boldsymbol{X}\right)$. Here, $\phi_{acc} \in \{0,1\}$ checks whether the label $y$ matches with the positioning of $\boldsymbol{X}$ w.r.t. the classifier of $\phi$, and $\phi_{mar}\left(\boldsymbol{X}\right) \in \mathbb{R}$ penalizes the margin, i.e. distance of $\boldsymbol{X}$ from the classifier's border. Now, let $\psi \subseteq \mathbf{\Phi}$ correspond to a subset of classifiers that classify all the data correctly ($\mathbb{E}_{P_0}\phi_{acc} = 0$), but have different expected margins. Also, assume $\phi^*$ (the minimum loss associated with the optimal classifier) is also inside $\psi$. Then, some simple calculations reveal that for every $\phi \in \psi$ and any $\lambda$ we have $\rho_\lambda\left(\phi\right) = 0$ (C.6) and thus $\Lambda\left(\psi\right) = -\infty$ (C.10). Also, we have $\Gamma\left(\psi; \lambda\right) \geq 0$ (C.9), again for any $\lambda$, while $\mathrm{GAP}\left(\psi\right)$ (C.9) is strictly positive for any non-trivial $\mathbf{\Phi}$ (recall that $\phi^* \in \psi$). Considering the fact that we can have $\zeta = O\left(n^{-1/2}\right)$ according to Theorem 3, then for a sufficiently large $n$, $\mathrm{MSR}_{\mathbf{\Phi},P_0}\left(\lambda, O\left(n^{-1/2}\right)\right)$ becomes zero for any $\lambda \in \mathbb{R} \cup \pm\infty$. This result is in full agreement with the previous bounds that are specifically derived for generic learnability of statistical models when non-overlapping cluster assumption holds (For absolute learnability, at least one data point with a label is needed to decide which cluster is which).

–(**minor comments**) 1. We have rephrased the sentences to avoid any possible confusions. 2. Yes, $n = n_\mathrm{l} + n_\mathrm{ul}$. Reviewer is correct and notations w.r.t. $\boldsymbol{D}$ will be corrected. 5. The model used in our experiments is a deep neural network whose structure is completely explained in the supplementary document. Unfortunately, we cannot give more info in the main text due to the page limit. 3.4.6. We will correct all the grammatical mistakes, and also update the references.

[Meta-Review · NeurIPS 2019]

The present paper proposes a method for detecting adversarial examples after the training phase. The idea is to compare the feature attribution map statistics of natural and adversarial examples. To obtain such result, this paper unifies two major learning frameworks: Semi-Supervised Learning and Distributionally Robust Learning. Also, the author proposes a new complexity measure that is an adversarial extension of a Rademacher complexity and its semi-supervised analogue. Namely this measure is related to the “need of supervision”. Based on this theoretical approach, the paper also proposed a new algorithm. The later comes with convergence guarantees. The paper is interesting and addresses an important problem for the machine learning community. Most of the reviewers agree on the good level of interest of the proposed result and to the fact that the paper is quite well written even if they would have like to see more explanations related to the proposed theoretical concepts. I strongly suggest to include in the camera ready, the example the authors provides in the rebuttal. Note that there is a code segment in the supplementary file, which includes a name that is likely to be one of the authors. This has been detected by one of the reviewer. However, as from the point of view of the reviewers, this had no effect on the scoring, and since this is, on my opinion a non-intentional error, I decide to not take this fact into account in my final decision.